# No Prompt Left Behind: Exploiting Zero-Variance Prompts in LLM Reinforcement Learning via Entropy-Guided Advantage Shaping

**Thanh-Long V. Le[1], Myeongho Jeon[2], Kim Vu[1], Viet Lai[3], Eunho Yang[1]**
[1] KAIST    [2] EPFL    [3] Adobe Research

## Abstract

Reinforcement Learning with Verifiable Rewards (RLVR) is a powerful framework for improving the reasoning abilities of Large Language Models (LLMs). However, current methods such as GRPO rely only on problems where the model responses to the same input differ in correctness, while ignoring those where all responses receive the same reward — so-called *zero-variance prompts*. In this work, we argue that such prompts are not useless but can, in fact, provide meaningful feedback for policy optimization. To this end, we introduce *Reinforcement Learning with Zero-Variance Prompts* (**RL-ZVP**), a novel algorithm that extract learning signals from zero-variance prompts. RL-ZVP directly rewards correctness and penalizes errors even without contrasting responses, modulating feedback with token-level characteristics to preserve informative, nuanced signals. Across six math reasoning benchmarks, RL-ZVP achieves significant improvements of up to 8.61 points in accuracy and 7.77 points in pass rate over GRPO, while consistently outperforming other baselines that filter out zero-variance prompts. These results highlight the untapped potential of learning from zero-variance prompts in RLVR. The project page is available at `https://bltnynk.github.io/publications/rl-zvp/`.

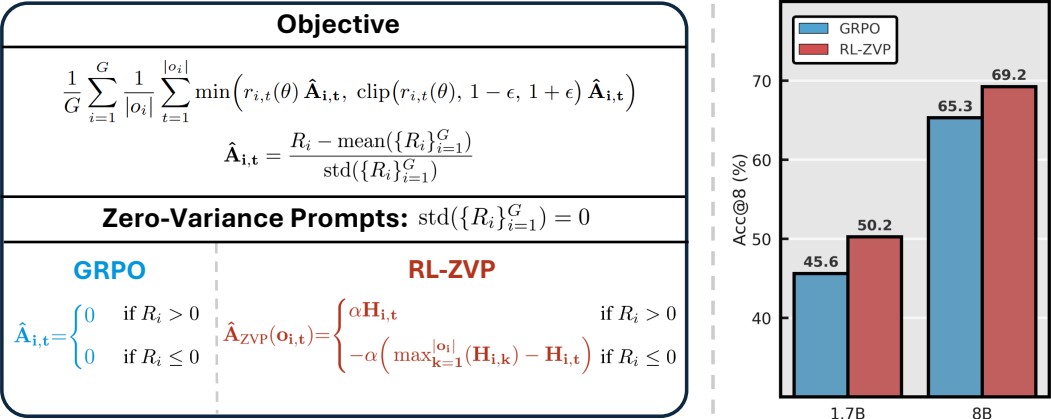

Figure 1: **Left**: RL-ZVP employs an entropy-guided advantage shaping formula to extract learning signals from zero-variance prompts. For non-zero-variance prompts, it reverts to the standard GRPO formulation. **Right**: RL-ZVP demonstrates significantly higher average accuracy than GRPO on both Qwen3-1.7B-Base and Qwen3-8B-Base across six math reasoning benchmarks.

Correspondence to: `thanhlong0780@kaist.ac.kr`, `daclai@adobe.com`

# 1 INTRODUCTION

Recent frontier models such as OpenAI o1 (Jaech et al., 2024), Claude 3.7 Sonnet (Anthropic, 2025) DeepSeek-R1 (Guo et al., 2025), and Kimi k1.5 (Team et al., 2025) provide compelling proof of concept for leveraging long chain of thought (Wei et al., 2022; Kojima et al., 2022) to enhance the reasoning capabilities of Large Language Models (LLMs). Following these advances, Reinforcement Learning with Verifiable Rewards (RLVR) techniques such as Group Relative Policy Optimization (GRPO) (Shao et al., 2024) and Dynamic Sampling Policy Optimization (DAPO) (Yu et al., 2025) have emerged as the de facto standard for strengthening LLM reasoning. In RLVR, the policy model is trained by sampling multiple responses (rollouts) to a prompt and then reinforcing correct answers while penalizing incorrect ones. While this approach is effective when responses for the same prompt contains both right and wrong responses, it does not account for scenarios when a group

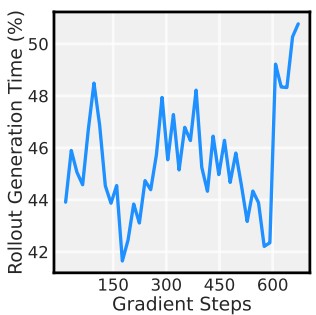

Figure 2: **Rollout generation overhead** as a percentage of total time of the training step.

of rollouts are entirely correct or entirely incorrect (with identical rewards), referred to as *zero-variance prompts* (Zheng et al., 2025). In such situations, the normalized rewards (*i.e.*, advantage values) collapse to zero, resulting in no training signal, rendering all rollouts for those prompts uninformative. While it may seem conceptually reasonable to ignore these prompts, doing so remains problematic because rollout generation itself is a major bottleneck in RL training (Zhong et al., 2025; Noukhovitch et al., 2025; Sheng et al., 2025). Our preliminary experiments show that response sampling typically takes up about 50% of the total time per training step, which emphasizes the significant throughput overhead introduced by rollouts (Figure 2).

Recent methods acknowledge the inefficiency of zero-variance prompts and attempt to filter them out, either pre-rollout (Zheng et al., 2025; Qu et al., 2026) or post-rollout (Yu et al., 2025). However, to the best of our knowledge, **no prior work has investigated whether zero-variance prompts themselves can serve as a valuable source of learning signals for policy optimization.**

In this paper, we challenge the prevailing consensus regarding zero-variance prompts and the common practice of filtering them out. We argue that, with an appropriate strategy, zero-variance prompts can still provide meaningful learning signals that benefit policy optimization during LLM training. We begin by analyzing the objective formulation of GRPO and identify the *advantage vanishing* issue that prevents zero-variance prompts from contributing to learning. Specifically, GRPO imposes implicit constraints on the policy optimization process: for a given prompt, the policy model is rewarded for correct responses only when at least one incorrect response exists within the group, and penalized for incorrect responses only when there is at least one correct response in the same group. Based on this observation, we develop an intuition for the high-level role of advantage in guiding token-level optimization, and how its interaction with response correctness influences the learning dynamics of language models.

Building on these insights, we introduce **RL-ZVP** (*Reinforcement Learning with Zero-Variance Prompts*), a novel algorithm that extracts useful learning signals from zero-variance prompts to improve the effectiveness of RL training. Our approach rests on two key intuitions: (i) the policy model should still be rewarded for correct answers even if no incorrect answers are present in the same group (and penalized analogously), and (ii) the degree of reward or penalty should be determined by the characteristics of sampled tokens, inspired by recent advances in reward shaping (Wang et al., 2025; Devidze et al., 2022; Ng et al., 1999). Concretely, we formulate the advantage for zero-variance prompts around two properties: *direction*, which aligns with the correctness of the response; and *magnitude*, which is quantified by the entropy value of each token within the response (Figure 1).

Experiments on six math reasoning benchmarks across two model scales — Qwen3-1.7B-Base and Qwen-8B-Base (Yang et al., 2025) — show that RL-ZVP significantly outperforms GRPO, with an average improvement of **4.00 points** in accuracy (Acc@8) and **4.28 points** in pass rate (Pass@8). Notably, RL-ZVP achieves gains of up to **8.66 points** in Acc@8 (on AIME25) and **7.77 points** in Pass@8 (on OlympiadBench). RL-ZVP also consistently surpasses other baselines that filter out zero-variance prompts, even under unfavorable settings where these baselines are given $3\times$–$5\times$

more rollouts. Moreover, across both scales, RL-ZVP demonstrates a more stable and consistent learning process. These results establish zero-variance prompts as a valuable resource for learning signals and open a new perspective on how to utilize them, rather than discarding them, in RLVR.

## 2 PRELIMINARY: GROUP RELATIVE POLICY OPTIMIZATION

Group Relative Policy Optimization (GRPO) (Shao et al., 2024) is a variant of Proximal Policy Optimization (PPO) (Schulman et al., 2017) tailored for LLM post-training. Given a question–answer pair $(q, a)$ from a dataset $\mathcal{D}$, a group of $G$ responses (rollouts) $\{o_i\}_{i=1}^G$ is sampled from the old policy $\pi_{\theta_{\text{old}}}$ (typically the policy from the previous training iteration). Each response is compared against the ground-truth answer $a$ to produce final rewards $\{R_i\}_{i=1}^G$. Instead of relying on a costly value model like PPO, GRPO normalizes these rewards to compute the advantage as:

$$\hat{A}_{i,t} = \frac{R_i - \text{mean}(\{R_i\}_{i=1}^G)}{\text{std}(\{R_i\}_{i=1}^G)}, \tag{1}$$

where $t$ denotes token position in $o_i$. Note that the advantage $\hat{A}_i$ is computed at the response level and subsequently broadcast to all tokens in the response, *i.e.*, $\hat{A}_{i,1} = \hat{A}_{i,2} = \cdots = \hat{A}_{i,G} = \hat{A}_i$. The policy objective[1] for a single prompt is then defined as:

$$\mathcal{J}_{\text{GRPO}}\Big(\theta \mid q, a, \{o_i\}_{i=1}^G\Big) = \frac{1}{G} \sum_{i=1}^G \frac{1}{|o_i|} \sum_{t=1}^{|o_i|} \min\Big(r_{i,t}(\theta)\,\hat{A}_{i,t},\ \text{clip}\big(r_{i,t}(\theta),\, 1-\epsilon,\, 1+\epsilon\big)\,\hat{A}_{i,t}\Big), \tag{2}$$

$$\text{where}\ \ r_{i,t}(\theta) = \frac{\pi_\theta(o_{i,t} \mid q,\, o_{i,<t})}{\pi_{\theta_{\text{old}}}(o_{i,t} \mid q,\, o_{i,<t})}.$$

Here, $r_{i,t}(\theta)$ represents the importance sampling ratio that adjusts the objective to account for the off-policy distribution of the training batch. The final GRPO objective for the whole dataset is:

$$\mathcal{J}_{\text{GRPO}}(\theta, \mathcal{D}) = \mathbb{E}_{(q,a)\sim\mathcal{D},\{o_i\}_{i=1}^G\sim\pi_{\theta_{\text{old}}}(\cdot|q)} \Big[\mathcal{J}_{\text{GRPO}}(\theta|q, a, \{o_i\}_{i=1}^G)\Big]. \tag{3}$$

## 3 REINFORCEMENT LEARNING WITH ZERO-VARIANCE PROMPTS

In this section, we introduce our algorithm, Reinforcement Learning with Zero-Variance Prompts. We begin by motivating the approach through the advantage vanishing problem in zero-variance prompts (Section 3.1). We then present our entropy-guided advantage formulation (Section 3.2) and conclude with the full training objective and algorithm (Section 3.3).

### 3.1 MOTIVATION: ADVANTAGE VANISHING IN ZERO-VARIANCE PROMPTS

During GRPO training, the policy model may encounter questions where all sampled responses in a group are either entirely correct or entirely incorrect, resulting in identical rewards. Following Zheng et al. (2025), we call these cases *zero-variance prompts*. They typically arise in two situations: when the dataset is either (i) too difficult or (ii) too easy for the model. From a training perspective, zero-variance prompts appear most frequently (i) at the beginning of training, when the model often fails to solve problems even once across multiple responses, and (ii) near convergence, when the model has become strong enough to solve many questions entirely.

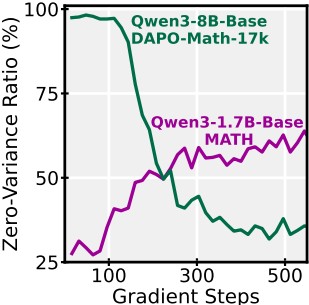

Figure 3: **The ratio of zero-variance prompts** across two experimental scales.

To understand the prevalence of zero-variance prompts, we examine two settings: Qwen3-1.7B-Base trained on MATH, and Qwen3-8B-Base trained on DAPO-Math-17k. We found that zero-variance prompts account for roughly 30% to 99% of each batch during training (Figure 3). This high proportion suggests that if these prompts could be leveraged effectively, they potentially represent an untapped source of useful learning signal for the policy model.

---

[1]We omit the KL penalty here for clarity.

**Remark 1** (Advantage Vanishing). *In Equation (1) and (2), if all responses within a group receive identical rewards, their advantages degenerate to zero. In this case, the GRPO objective reduces to $\mathcal{J}_{GRPO}\big(\theta \mid q_{zvp}, a, \{o_i\}_{i=1}^{G}\big) = 0$, effectively discarding all the potential learning signal.*

From Equation (2), taking the gradient of both sides[2] gives:

$$\nabla_\theta \mathcal{J}_{\text{GRPO}}(q, a, \{o_i\}_{i=1}^{G}) = \frac{1}{G} \sum_{i=1}^{G} \frac{1}{|o_i|} \sum_{t=1}^{|o_i|} r_{i,t}(\theta) \, \hat{A}_{i,t} \nabla_\theta \log \pi_\theta\big(o_{i,t} \mid q, o_{i,<t}\big). \tag{4}$$

From this expression, the gradient of the GRPO objective with respect to the policy parameters can be interpreted as a weighted sum of the log-probability gradients for all tokens in a response. Intuitively, GRPO raises the probabilities of tokens in correct responses while lowering those of tokens in incorrect ones. Under GRPO advantage formulation, however, if we ignore the importance-sampling ratio, each token in a response receives the same weight, regardless of its role or informativeness.

**Remark 2** (The Role of Advantage). *The sign of the advantage $\hat{A}_{i,t}$ determines the **direction** of the gradient update for each token since the importance sampling ratio $r_{i,t}$ in Equation (4) is strictly positive, while its absolute value $|\hat{A}_{i,t}|$ governs the **magnitude** of the update in conjunction with $r_{i,t}$.*

Recently, several methods have excluded zero-variance prompts from RL training (Yu et al., 2025; Zheng et al., 2025; Cheng et al., 2026), viewing them merely as a way to reduce rollout cost and implicitly assuming they offer no value for policy optimization. We challenge this assumption and argue that, when handled appropriately, ***zero-variance prompts could provide valuable learning signals for RL training.***

### 3.2 ADVANTAGE FORMULATION

We propose ***Reinforcement Learning with Zero-Variance Prompts*** (RL-ZVP), a novel algorithm designed to exploit zero-variance prompts. The key intuition is to address the limitation noted in Remark 1: the policy should be rewarded or penalized based solely on the correctness of its responses, even when no contrasting answers are present. RL-ZVP behaves identically to GRPO on non-zero-variance prompts, but adopts a different strategy for zero-variance ones, thereby generalizing GRPO. We first classify zero-variance prompts with all correct responses as *positive prompts* and those with all incorrect responses as *negative prompts*. Based on Remark 2, we frame advantage shaping as developing appropriate scaling factors, with a focus on two properties: *direction* and *magnitude*.

**Advantage direction.** The sign of the advantage is straightforward: positive $(+)$ for positive prompts and negative $(-)$ for negative ones. Intuitively, when all sampled responses are correct, the model should increase their probabilities, reinforcing certainty and encouraging exploitation. Conversely, when all responses are incorrect, their probabilities should be reduced, discouraging repeated mistakes and promoting exploration of alternative sampling paths.

**Advantage magnitude.** Inspired by recent works that incorporate token-level signals to augment verifiable rewards (Cui et al., 2025; Fang et al., 2025; Agarwal et al., 2025; Deng et al., 2025; Lei et al., 2025), we leverage token entropy in the formulation of the advantage magnitude for zero-variance prompts. The key idea is to scale each token's gradient update in proportion to its entropy. For correct responses, high-entropy tokens — typically those crucial for reasoning, such as connectors or branching points(Wang et al., 2025) — receive larger updates than semantically trivial tokens. This encourages reasoning behaviors such as reflection and verification (Guo et al., 2025), while avoiding over-exploitation of trivial text completion patterns. For incorrect responses, the probabilities of all tokens are reduced, but high-entropy tokens (that initiate reasoning paths and behaviors) are penalized less severely, preserving flexibility for the model to revisit and explore those reasoning paths in future iterations.

---

[2]We omit the clipping factor before gradient derivation for clarity.

**Algorithm 1:** Reinforcement Learning with Zero-Variance Prompts

---

**Input:** initial policy model $\pi_\theta$; training dataset $\mathcal{D}$;

**for** *iteration* $= 1, \ldots, M$ **do**

    Update the previous policy $\pi_{\theta_{\text{old}}} \leftarrow \pi_\theta$;

    Sample (w/o replacement) a prompt batch $\mathcal{B} \subset \mathcal{D}$;

    **foreach** $(q, a) \in \mathcal{B}$ **do**

        Sample $G$ outputs $\{o_i\}_{i=1}^G \sim \pi_{\theta_{\text{old}}}(\cdot \mid q)$;

        Compute rewards $\{R_i\}_{i=1}^G$ by verifying each $o_i$ against $a$;

    **for** *gradient_step* $= 1, \ldots, \mu$ **do**

        Sample (w/o replacement) a mini-batch $\mathcal{B}_{\text{mini}} \subset \mathcal{B}$;

        **foreach** $(q, a, \{R_i\}_{i=1}^G) \in \mathcal{B}_{mini}$ **do**

            **if** $\text{std}(\{R_i\}_{i=1}^G) > 0$ **then** compute advantages $\{\hat{A}_{i,t}\}$ using Equation (1);

            **else** compute advantages $\{\hat{A}_{\text{ZVP}}(o_{i,t})\}$ using Equation (5);

        Update the policy model $\pi_\theta$ by maximizing the RL-ZVP objective (6);

**Output:** final policy model $\pi_\theta$

---

**Advantage formula.** Combining these two design choices, our final token-level advantage formulation for zero-variance prompts, replacing the zero assigned in GRPO, is:

$$\hat{A}_{\text{ZVP}}(o_{i,t}) = \begin{cases} \alpha H_{i,t}, & \text{if } \text{std}(\{R_i\}_{i=1}^G) = 0 \ \wedge \ R_i > 0, \\ -\alpha\Big( \max_{k=1}^{|o_i|}(H_{i,k}) - H_{i,t} \Big), & \text{if } \text{std}(\{R_i\}_{i=1}^G) = 0 \ \wedge \ R_i \leq 0, \end{cases} \tag{5}$$

$$\text{with} \quad H_{i,t} = -\sum_{j=1}^{|V|} \pi_\theta(v_j \mid q, o_{i,<t}) \log\big[\pi_\theta(v_j \mid q, o_{i,<t})\big],$$

where $\alpha$ denotes the scaling factor hyperparameter, and $H_{i,t}$[3] is the entropy of each token $o_{i,t}$ in the response $o_i$ given the policy model's vocabulary set $V = \{v_1, v_2, \cdots\}$. The term $\max_{k=1}^{|o_i|}(H_{i,k})$ reverts the effect of token entropy in negative prompts to align with our intuition: tokens with higher entropy incur smaller penalties, while low-entropy tokens are penalized more.

### 3.3 FINAL OBJECTIVE

Let $\mathcal{D}_{\text{ZVP}}$ be the set of zero-variance prompts encountered during training ($\mathcal{D}_{\text{ZVP}} \subset \mathcal{D}$). Incorporating our advantage shaping for zero-variance prompts, the RL-ZVP objective becomes:

$$\mathcal{J}_{\text{RL-ZVP}}(\theta|q, a, \{o_i\}_{i=1}^G) = \begin{cases} \frac{1}{G}\sum_{i=1}^G \frac{1}{|o_i|}\sum_{t=1}^{|o_i|} \min\Big(r_{i,t}(\theta)\,\hat{A}_{\text{ZVP}}(o_{i,t}), \\ \qquad\quad \text{clip}(r_{i,t}(\theta), 1-\epsilon, 1+\epsilon)\,\hat{A}_{\text{ZVP}}(o_{i,t})\Big), & \text{if } q \in \mathcal{D}_{\text{ZVP}}, \\ \mathcal{J}_{\text{GRPO}}(\theta|q, a, \{o_i\}_{i=1}^G), & \text{otherwise.} \end{cases} \tag{6}$$

**Remark 3** (Relationship to GRPO). *RL-ZVP does not introduce any new loss term beyond the GRPO objective. It simply reformulates the special case where the prompt is zero-variance. In fact, the GRPO objective from Equation (2) can be equivalently expressed in a branching form:*

$$\mathcal{J}_{GRPO}(\theta \mid q, a, \{o_i\}_{i=1}^G) = \begin{cases} 0, & \text{if } q \in \mathcal{D}_{\text{ZVP}}, \\ \mathcal{J}_{GRPO}(\theta \mid q, a, \{o_i\}_{i=1}^G), & \text{otherwise.} \end{cases} \tag{7}$$

*In this regard, our formulation serves as a generalization of GRPO—reducing exactly to GRPO when prompts are non-zero-variance, while resulting in a non-trivial update for zero-variance prompts rather than collapsing to zero.*

---

[3]In the implementation, we detach every occurrence of $H_{i,t}$ from the computational graph so that it acts as a scalar factor rather than a differentiable parameter.

## 4 EXPERIMENTS

In this section, we first outline our experimental setup, including models, datasets, and implementation details (Section 4.1). We then present validation results and analysis, comparing RL-ZVP with GRPO and prompt filtering baselines (Section 4.2). Finally, we conduct ablation studies to assess the contribution of each sub-component in RL-ZVP (Section 4.3).

### 4.1 EXPERIMENTAL SETTINGS

**Models & Datasets.** We conduct experiments on both Qwen3-1.7B-Base and Qwen3-8B-Base (Yang et al., 2025; Liu et al., 2026) to assess performance across model scales. For Qwen3-1.7B-Base, we train on the MATH dataset (Hendrycks et al., 2021; Lightman et al., 2024), which contains 7.5k problems. For Qwen3-8B-Base, we adopt the more challenging DAPO-Math-17k dataset (Yu et al., 2025) to better exploit the capacity of larger models. Within the scope of this work, we refer to these two setups as *small scale* and *large scale*, respectively. We then evaluate the trained models on six standard math reasoning benchmarks: Minerva, AMC23, MATH (Hendrycks et al., 2021), AIME24, AIME25, and OlympiadBench (He et al., 2024). For each question, we generate eight responses and report both the accuracy (Acc@8) and the pass rate (Pass@8). To further evaluate the generalization of RL-ZVP, we also report additional experimental results in Appendix B.

**Baselines.** Since RL-ZVP is the first method that attempts to explicitly leverage zero-variance prompts, we compare it against GRPO (Shao et al., 2024) and two representative methods that filter out zero-variance prompts: GRPO with Dynamic Sampling (GRPO-DS) (Yu et al., 2025) and GRESO (Zheng et al., 2025). GRPO-DS repeatedly samples questions, generates group of responses, discards zero-variance cases, and constructs batches containing only non-zero-variance prompts, ensuring that every prompt contributes to the gradient signal. In contrast, GRESO predicts and skips zero-variance prompts without responses sampling, using a probabilistic filtering strategy based on historical training dynamics. For GRPO-DS and GRESO, we report results under two settings: *(r)* using the *same number of rollouts* as RL-ZVP for a fair comparison, and *(g)* using the *same number of gradient steps*. The latter favors the prompt filtering baselines, as it allows roughly $3\times$–$5\times$ more rollouts than RL-ZVP and GRPO (Table 1).

Table 1: **Number of rollouts and gradient steps of RL-ZVP compared to the baselines.** For GRPO-DS-r and GRESO-r on the large scale, the policy initially solves only $\sim$2% of problems per training batch. This leads to extremely poor data efficiency, requiring over-sampling of $\sim$50$\times$ to form a full non-zero-variance batch for each gradient step. As a result, these methods achieve far fewer gradient updates than GRPO and RL-ZVP, despite using nearly 1.5$\times$ more rollouts. This highlights the severe inefficiency of dynamic sampling on challenging training data.

| Method | Small Scale | | | Large Scale | | |
|---|---|---|---|---|---|---|
| | #Rollouts | Rollouts ($\times$GRPO) | Grad Steps | #Rollouts | Rollouts ($\times$GRPO) | Grad Steps |
| GRPO | 172,032 | 1.00$\times$ | 672 | 540,672 | 1.00$\times$ | 1,056 |
| GRPO-DS-r | 172,032 | 1.00$\times$ | 320 | 827,392 | 1.53$\times$ | 32 |
| GRPO-DS-g | 421,888 | 2.45$\times$ | 672 | 2,859,008 | 5.29$\times$ | 1,056 |
| GRESO-r | 173,600 | 1.01$\times$ | 320 | 815,232 | 1.51$\times$ | 32 |
| GRESO-g | 272,030 | 1.58$\times$ | 672 | 2,159,104 | 3.99$\times$ | 1,056 |
| **RL-ZVP** | 172,032 | 1.00$\times$ | 672 | 540,672 | 1.00$\times$ | 1,056 |

**Implementation Details.** All experiments are implemented using the `verl` framework (Sheng et al., 2025), following the standard hyperparameter settings from recent works (Yu et al., 2025; Zheng et al., 2025). We train with a batch size of 512 and a mini-batch size of 32, resulting in 16 gradient steps per training iteration. Full implementation details are provided in Appendix A.

### 4.2 RESULTS & ANALYSIS

**RL-ZVP achieves improvements of up to 8.61 points in accuracy and 7.77 points in pass rate over GRPO.** We present our main results in Table 2 and Figure 4. Across six benchmarks, RL-ZVP

Table 2: **Detailed evaluation results on six math reasoning benchmarks.** The best and second best performance across all settings are **bold** and underscored, respectively. Under a fair setup of equal rollout budget, RL-ZVP achieves the best performance across all datasets, heavily outperforming GRPO. (*) Even in the unfavorable setting where Dynamic Sampling baselines use $3\times-5\times$ more rollouts, RL-ZVP still outperforms them on the majority of benchmarks.

| Method | Minerva | | AMC23 | | MATH500 | | AIME24 | | AIME25 | | OlympiadBench | |
|---|---|---|---|---|---|---|---|---|---|---|---|---|
| | Acc@8 | Pass@8 | Acc@8 | Pass@8 | Acc@8 | Pass@8 | Acc@8 | Pass@8 | Acc@8 | Pass@8 | Acc@8 | Pass@8 |
| **Qwen3-1.7B-Base** trained with **MATH** (*small scale*) | | | | | | | | | | | | |
| GRPO | 29.09 | 43.19 | 42.19 | 72.00 | 69.09 | 84.05 | 8.75 | 17.24 | 4.17 | 6.67 | 33.20 | 50.52 |
| GRPO-DS-r | 29.96 | 44.02 | 42.50 | 73.84 | 68.31 | 83.04 | 7.50 | 17.41 | 2.92 | 8.73 | 32.67 | 49.89 |
| *GRPO-DS-g** | 29.96 | 44.02 | 46.25 | 73.84 | 70.72 | 84.61 | 7.50 | **25.77** | **7.50** | 12.95 | **35.68** | 53.03 |
| GRESO-r | 27.99 | 43.39 | 40.00 | 70.25 | 67.53 | 83.07 | 7.08 | 18.48 | 3.75 | 12.89 | 31.30 | 49.13 |
| *GRESO-g** | **30.19** | 43.39 | 45.00 | 73.86 | 69.77 | 84.77 | 9.58 | 19.51 | 5.00 | 11.00 | 34.68 | 52.43 |
| **RL-ZVP (Ours)** | 29.96 | **44.76** | **48.75** | **77.19** | 70.98 | **87.35** | 12.50 | 23.99 | 6.25 | **13.23** | 35.11 | **54.84** |
| $\Delta$(w.r.t GRPO) | +0.87 | +1.57 | +6.56 | +5.19 | +1.89 | +3.3 | +3.75 | +6.75 | +2.08 | +6.56 | +1.91 | +4.32 |
| **Qwen3-8B-Base** trained with **DAPO-Math-17k** (*large scale*) | | | | | | | | | | | | |
| GRPO | 44.99 | 58.42 | 73.75 | 92.48 | 83.00 | 92.40 | 28.33 | 50.64 | 20.00 | 31.59 | 49.59 | 66.20 |
| GRPO-DS-r | 29.96 | 50.85 | 47.19 | 76.70 | 68.20 | 87.26 | 12.08 | 31.33 | 9.17 | 22.02 | 43.88 | 58.16 |
| *GRPO-DS-g** | 44.76 | **59.60** | 76.25 | 92.43 | 84.98 | 92.42 | 27.08 | 42.72 | 19.58 | 35.93 | 44.83 | 60.42 |
| GRESO-r | 30.74 | 50.53 | 49.38 | 80.53 | 67.40 | 87.79 | 12.08 | 29.31 | 10.83 | 26.26 | 44.92 | 59.66 |
| *GRESO-g** | 45.50 | 58.69 | 78.13 | 92.90 | 83.23 | 92.24 | 25.00 | 41.80 | 17.92 | 32.12 | 44.14 | 59.79 |
| **RL-ZVP (Ours)** | **45.96** | 58.83 | **78.75** | **94.18** | **89.73** | **96.13** | **33.33** | **54.66** | **24.58** | **39.36** | **58.20** | **72.28** |
| $\Delta$(w.r.t GRPO) | +0.97 | +0.41 | +5.00 | +1.70 | +6.73 | +3.73 | +5.00 | +4.02 | +4.58 | +7.77 | +8.61 | +6.08 |

consistently delivers significant gains over GRPO, ranging from 0.87 to 6.75 points on Qwen3-1.7B-Base and from 0.41 to 8.61 points on Qwen3-8B-Base. On small scale, the average improvements are 2.84 points in Acc@8 and 4.62 points in Pass@8, while on large scale, the average gains are 5.15 points in Acc@8 and 3.95 points in Pass@8. These results indicate that leveraging zero-variance prompts provides strong and informative learning signals that substantially enhance reasoning performance, with improvements generalizing across both model scales and datasets.

**RL-ZVP consistently outperforms prompt-filtering baselines, even under disadvantageous settings.** Under the fair setting with the same rollout budget, RL-ZVP substantially surpasses both GRPO-DS-r and GRESO-r across all benchmarks, achieving average gains of 3.28 points in Acc@8 and 4.03 points in Pass@8 on the small scale, 19.20 points in Acc@8 and 13.56 points in Pass@8 on the large scale (Figure 4). Even in the less favorable setting with the same number of gradient steps — where the baselines require $3\times-5\times$ more rollouts (Table 1) — RL-ZVP still heavily outperforms GRPO-DS-g and GRESO-g on the majority of benchmarks, with average improvements (over their best performance) of 0.99 points in Acc@8 and 1.19 points in Pass@8 on the small scale, and 5.51 points in Acc@8 and 5.32 points in Pass@8 on the large scale. These results not only reaffirm the importance of learning signals from zero-variance prompts, but also provide a new insight: lever-

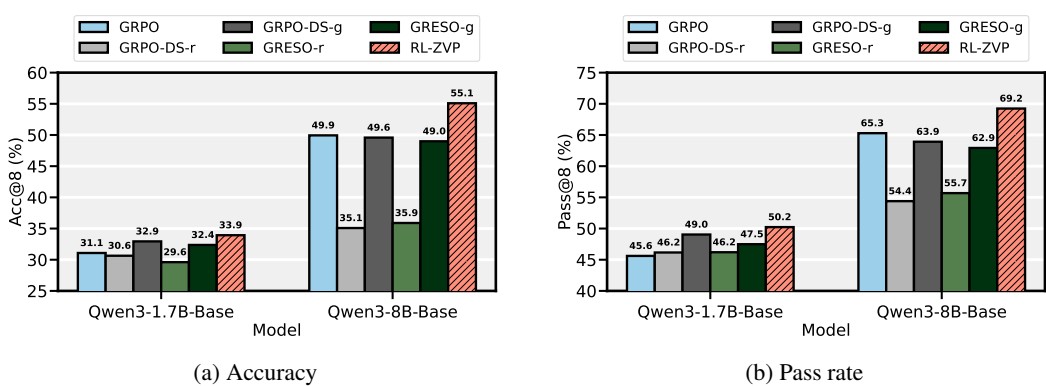

(a) Accuracy          (b) Pass rate

Figure 4: Average **accuracy (a)** and **pass rate (b)** on six math reasoning benchmarks. RL-ZVP consistently delivers the strongest performance among all baselines.

aging such prompts brings greater benefits than discarding them, directly challenging the prevailing practice adopted in recent works.

(a) Acc@8 (small scale)  (b) Entropy (small scale)  (c) Response Length (small scale)

(d) Acc@8 (large scale)  (e) Entropy (large scale)  (f) Response Length (large scale)

Figure 5: **Validation accuracy and training dynamics at different experiment scales.** Each row shows Acc@8, entropy, and response length during training for Qwen3-1.7B-Base (**top**) and Qwen3-8B-Base (**bottom**). RL-ZVP exhibits more consistent and stable trends than GRPO.

**RL-ZVP demonstrates a more stable and consistent learning process.** Figure 5 present the validation accuracy and learning dynamics across benchmarks of RL-ZVP and GRPO as training progresses. On 1.7B scale, RL-ZVP initially shows competitive results with GRPO, but after roughly 500 gradient steps, it achieves a clear performance boost, creating a significant performance gap. On 8B scale, the difference is even more pronounced: RL-ZVP exhibits a smooth and steadily improving curve, while GRPO suffers a sharp drop during the first 200 gradient steps, then recovers and but eventually underperforms RL-ZVP. This highlights RL-ZVP's superior effectiveness and stability, which avoids the performance collapses observed at the start of training with GRPO on large scale.

We attribute this robustness to the quantity and quality of learning signals each method receives at different training stages. In small scale experiments, the number of zero-variance prompts increases as training progresses (Figure 3). The model improves and produces more positive prompts (all-correct responses), which RL-ZVP can leverage as useful signals, while GRPO discards them. This leads to a widening gap in later stages of training. For large-scale experiments, the situation is reversed at the start of training: the model initially solves fewer than 5% of problems per batch (Figure 3), leaving GRPO with sparse and noisy feedback. In contrast, RL-ZVP can still exploit the abundance of negative prompts (all-incorrect responses), obtaining denser and thereby resulting in greater training stability in the early stages. Although the proportion of zero-variance prompts decreases as training continues, it remains non-negligible, sustaining a huge advantage for RL-ZVP over GRPO. Overall, across both scales, RL-ZVP consistently benefits from the additional signals provided by zero-variance prompts, ensuring stronger and more stable performance than GRPO.

**RL-ZVP encourages longer and more confident reasoning across model scale.** As training progresses, RL-ZVP steadily reduces entropy while producing increasingly longer responses. Both trends align with significant gains in validation results (Figure 5). These patterns, along with the qualitative demonstrations (Appendix F), suggest two complementary effects related to the policy model: (i) enhanced capabilities due to the emergence of long-form reasoning behaviors — the so-called *aha-moment* (Guo et al., 2025; Jin et al., 2024) — and (ii) greater confidence in exploratory decisions, as reflected in the low entropy (Cheng et al., 2026; Kadavath et al., 2022).

On the small scale, models trained with RL-ZVP show a significant increase in response length over training, indicating the ability to generate extensively long chains of reasoning that greatly improves its reasoning abilities (Jin et al., 2024; Guo et al., 2025). In contrast, GRPO-trained models exhibit minimal growth in response length, aligning with their bounded validation performance and potentially their inability to generate multi-step, well-structured solutions. On the large scale, entropy initially drops sharply but then stabilizes after roughly 300 gradient steps rather than collapsing.

Combined with the consistent growth in response length and accuracy, this rules out entropy collapse and emphasizes RL-ZVP's ability to promote detailed and confident reasoning (Cui et al., 2025; Wang et al., 2024a).

**RL-ZVP facilitates more sophisticated reasoning even with small policy models.** In many cases, especially on 1.7B scale, GRPO-trained models produce simple and direct solutions, jumping straight into problem-solving with little structure. Even when they show glimpses of more complex behaviors such as reflection, these behaviors are shallow and disorganized. In contrast, models trained with RL-ZVP display much more sophisticated and systematic reasoning strategies: they restate and reflect on the key hypotheses, carefully interpret the clues, plan solutions step by step, organize their reasoning into major steps with detailed sub-solutions, and reasonably attempt alternative strategies when encountering dead ends. Illustrative examples are provided in Appendix F.

For instance, during exploration, GRPO-trained models often rely on shallow trial-and-error heuristics (*e.g.*, "*let's try another set of values*", "*let's try a different method*"), which resemble second-guessing and lack of direction. In contrast, models trained with RL-ZVP adopt more principled and comprehensive strategies (*e.g.*, "*let's double check*", "*assume symmetry to simplify the problem*", "*try to factor directly*", "*let's try to express the sides using the Law of Cosines*"), demonstrating focused and systematic exploration. Remarkably, such behaviors emerge even in models as small as 1.7B when trained with RL-ZVP, whereas GRPO fails to elicit them at this scale. On 8B scale, both GRPO and RL-ZVP produce systematic solutions with clear structure. However, GRPO outputs tend to be repetitive and rigid in tone, while RL-ZVP exhibits a more flexible and exploratory style of reasoning, reflecting a deeper ability to search for and evaluate alternative solution paths.

## 4.3 ABLATION STUDY

To better understand the contribution of each design choice in RL-ZVP, we conduct ablation experiments on our small scale setting — Qwen3-1.7B-Base trained with MATH.

**Impact of zero-variance type and entropy scaling.** We examine the contribution of three subcomponents in our advantage formulation for RL-ZVP: negative prompts, positive prompts, and the entropy scaling factor. Table 3 reports the performance when each component is removed, comparing against full RL-ZVP and GRPO. The results are consistent across benchmarks: full RL-ZVP achieves the best performance in all evaluations. Ignoring either negative or positive prompts yields comparable performance, confirming that both play similar roles in providing learning signals for the policy models. Importantly, removing the entropy scaling factor (*e.g.*, using only sample-level +1/-1 as the advantage) causes the most severe degradation. This emphasizes the importance of *advantage magnitude* design for zero-variance prompts learning signal: simply promoting correct answers and penalizing incorrect ones is insufficient. Rewards and penalties must also be weighted appropriately to be effective.

Table 3: **Ablation on the subcomponents of RL-ZVP**. Full RL-ZVP achieves the best average performance, while the second-best results mostly occur when a single component is removed.

| Method | Minerva | | AMC23 | | MATH500 | | Average | |
|---|---|---|---|---|---|---|---|---|
| | Acc@8 | Pass@8 | Acc@8 | Pass@8 | Acc@8 | Pass@8 | Acc@8 | Pass@8 |
| **RL-ZVP (Ours)** | **29.96** | **44.76** | **48.75** | **77.19** | **70.98** | **87.35** | **49.90** | **69.77** |
| - w/o negative prompts | 29.92 | 43.87 | 44.69 | 70.51 | 68.64 | 84.40 | 47.75 | 66.26 |
| - w/o positive prompts | 29.50 | 43.88 | 41.56 | 70.40 | 70.70 | 85.81 | 47.25 | 66.70 |
| - w/o entropy scaling | 28.95 | 43.02 | 42.50 | 70.76 | 69.19 | 84.37 | 46.88 | 66.05 |
| GRPO | 29.09 | 43.19 | 42.19 | 72.00 | 69.09 | 84.05 | 46.79 | 66.41 |

**Impact of $\alpha$.** At the beginning of training, the top 20% of tokens with the highest entropy often take values between 2 and 10, which can result in very large advantages if left unscaled. To avoid exploding updates and potential training instability, we introduce $\alpha$ as the scaling factor in the advantage computation. Table 4 shows the performance of RL-ZVP under different $\alpha$ values. We observe that RL-ZVP achieves its best results with $\alpha = 0.10$ or $\alpha = 0.20$, while performance drops sharply at both $\alpha = 0.05$ and $\alpha = 0.30$. This demonstrates that $\alpha$ plays a critical role in balancing stability and effective learning: overly small values fail to sufficiently utilize learning signal from zero-variance prompts, while overly large values may cause unstable training with large gradient updates.

Table 4: **Validation results under different $\alpha$ values**. Both excessively small and excessively large values degrade performance. Across all experimental settings, we empirically find that $\alpha = 0.1$ yields the most consistent and best performance.

| Alpha | Minerva | | AMC23 | | Math500 | | Average | |
|---|---|---|---|---|---|---|---|---|
| | Acc@8 | Pass@8 | Acc@8 | Pass@8 | Acc@8 | Pass@8 | Acc@8 | Pass@8 |
| $\alpha = 0.05$ | 29.60 | 43.78 | 42.19 | 69.44 | 68.56 | 82.96 | 46.78 | 65.39 |
| $\alpha = 0.10$ | **29.96** | **44.76** | **48.75** | **77.19** | 70.98 | **87.35** | **49.90** | **69.77** |
| $\alpha = 0.20$ | 29.60 | 44.54 | 43.75 | 74.14 | **71.10** | 86.47 | 48.15 | 68.38 |
| $\alpha = 0.30$ | 28.68 | 42.61 | 41.56 | 71.22 | 67.20 | 83.46 | 45.81 | 65.76 |

## 5 RELATED WORK

**Reinforcement Learning for LLM Reasoning.** With the rise of test-time scaling and large reasoning models (Muennighoff et al., 2025; Guo et al., 2025), the post-training stage of large language models has increasingly shifted toward Reinforcement Learning with Verifiable Rewards (RLVR). However, these methods either discard zero-variance prompts implicitly (Shao et al., 2024; Liu et al., 2025) or explicitly (Yu et al., 2025; Zheng et al., 2025). This leads to wasted computation and reduced data efficiency during online training as rollouts remains a major bottleneck in RL training (Zhong et al., 2025; Noukhovitch et al., 2025). Our work addresses this gap by extracting useful signals from zero-variance prompts, thereby mitigating useless rollout costs.

**Online Prompt Selection.** Zero-variance prompts have been identified as a key weakness of GRPO in recent works (Yu et al., 2025; Zheng et al., 2025; Qu et al., 2026). DAPO addresses this issue with post-rollout oversampling: repeatedly sampling until a batch containing only non-zero-variance prompts is obtained (Yu et al., 2025). GRESO improves efficiency with a pre-rollout probabilistic filter, leveraging training history to predict whether a prompt is likely to be zero-variance (Zheng et al., 2025). While these approaches can be broadly viewed as online prompt selection methods, our work differs fundamentally: instead of filtering out zero-variance prompts, we propose extracting useful learning signals from them, thereby improving the effectiveness of RL training.

**Fine-grained Reward Shaping.** In contrast to traditional RL algorithms such as PPO (Schulman et al., 2017), recent RLVR methods remove the value model and compute sample-level advantages, broadcasting the same advantage to all tokens in a response. This coarse-grained approach has motivated research into fine-grained reward shaping, such as token-level adjustments based on token roles (Cui et al., 2025; Agarwal et al., 2025; Wang et al., 2025; Deng et al., 2025; Lei et al., 2025). However, these methods still suffer from advantage vanishing when faced with zero-variance prompts. Our work builds on this line of research by leveraging entropy as a mechanism to extract meaningful signals from zero-variance prompts, addressing this limitation directly.

## 6 CLOSING REMARKS

In this paper, we introduced RL-ZVP, a novel algorithm that extracts useful learning signals from zero-variance prompts to enhance policy optimization. Within these prompts, RL-ZVP directly rewards correct responses and penalizes incorrect ones, without requiring contrastive answers within the same group. Furthermore, RL-ZVP adopts a fine-grained optimization strategy, scaling updates by token entropy to elicit stronger reasoning behaviors in the policy model. Empirical results show that RL-ZVP delivers significant improvements, substantially outperforming GRPO and other strong baselines. To the best of our knowledge, this is the first work to demonstrate that zero-variance prompts can be exploited rather than discarded. We believe this perspective opens a new direction for RL training, with possibilities for larger gains through improved formulations in future research.

**Limitations & Future Directions.** Due to computational constraints, our experiments are limited to models up to 8B parameters; future work should investigate how RL-ZVP scales to larger models (*e.g.*, 14B or 32B). Furthermore, we only validate RL-ZVP on verifiable tasks with binary rewards; extending it to settings with graded or ambiguous feedback remains an open challenge. Finally, a promising direction is to generalize our advantage formulation beyond zero-variance prompts or to combine it with existing token-level reward shaping methods to further improve GRPO.

ACKNOWLEDGEMENT

We thank Brano Kveton, Nikos Vlassis, Yoonjeon Kim, Doohyuk Jang, Gyouk Chu, Minh-Hai Nguyen, and Tri Le for their valuable feedback and discussions. We are also grateful to Trung Bui for supporting compute allocation.

This work was supported by the National Research Foundation of Korea (NRF) grant (No. RS-2023-00209060, A Study on Optimization and Network Interpretation Method for Large-Scale Machine Learning) and the Institute of Information & Communications Technology Planning & Evaluation (IITP) grant (No. RS-2019-II190075, Artificial Intelligence Graduate School Program (KAIST); No. RS-2024-00457882, AI Research Hub Project) funded by the Korea government (MSIT). Myeongho Jeon was supported by the Sejong Science Fellowship of the National Research Foundation of Korea (NRF) (No. RS-2024-00421203) and the InnoCORE Program of the Ministry of Science and ICT (No. N10250156).

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

# APPENDIX

## A    FULL IMPLEMENTATION DETAILS

We present the full experimental setup details here. RL-ZVP and all baselines are implemented on top of the `verl` framework, ensuring that every experiment is conducted within the same pipeline. Training is performed on 4×RTX 3090 GPUs for Qwen3-1.7B-Base and 8×A100 GPUs for Qwen3-8B-Base.

For evaluation, we initially attempted to measure validation results after every training iteration (16 gradient steps), but this proved extremely inefficient given the size of the validation set (1534 problems with 8 responses each). Instead, on the small scale, we evaluate the model performance on Minerva, AMC23, and MATH after every iteration and report the best result, while for AIME24, AIME25, and OlympiadBench we use the final checkpoint. On the large scale, thanks to greater training efficiency on more powerful GPUs, we evaluate Minerva, AMC23, MATH, AIME24, and AIME25 after every iteration and report the best results, while for OlympiadBench we use only the final checkpoint.

The detailed hyperparameter settings are provided in Table 5. We mostly follow the standard hyperparameter settings from recent works (Yu et al., 2025; Cheng et al., 2026; Zheng et al., 2025). We use the clip-higher scheme with $\epsilon_{\text{high}} = 0.28$ and $\epsilon_{\text{low}} = 0.2$, apply token-level loss, and exclude the KL divergence term from the objective.

In addition to the common settings shared across methods, we also adopt the unique hyperparameter configurations of each baseline. For GRESO (Zheng et al., 2025), we follow their probabilistic filtering setup exactly: $\alpha_{\text{easy}} = \frac{1}{12}$, $\alpha_{\text{hard}} = \frac{1}{6}$, $p_{\text{easy}} = 0.5$, $p_{\text{hard}} = 0.5$, $\Delta p = 0.01$, and $\beta = 1.25$. For GRPO-DS, the only deviation from DAPO (Yu et al., 2025) is that we omit the *soft overlong punishment* to ensure a fair comparison with other methods, which also do not include this form of length-based reward shaping.

Table 5: **Full hyperparameter setting.**

| Hyperparameters | Qwen3-1.7B-Base + MATH | Qwen3-8B-Base + DAPO-Math-17k |
|---|---|---|
| Training batch size | 512 | 512 |
| Mini batch size | 32 | 32 |
| Samples per prompt | 8 | 16 |
| Max response length | 4096 | 8192 |
| Clip ratio | (0.20, 0.28) | (0.20, 0.28) |
| Scaling factor $\alpha$ | 0.10 | 0.10 |
| Training temperature | 1.0 | 1.0 |
| Training top_p | 1.0 | 1.0 |
| Validation temperature | 1.0 | 1.0 |
| Validation top_p | 0.7 | 0.7 |
| Number of epochs | 3 | 2 |
| Total gradient steps | 672 | 1056 |
| Optimizer | AdamW | AdamW |
| Learning rate | $3 \times 10^{-6}$ | $1 \times 10^{-6}$ |
| LR warmup steps | 10 | 10 |
| LR scheduler | cosine | constant |

## B    ADDITIONAL EXPERIMENTAL RESULTS

### B.1    ALTERNATIVE TRAINING DATA

To further validate the effectiveness of RL-ZVP on an alternative training dataset, we conduct experiments using a random 20k-prompt subset of NuminaMath-CoT (LI et al., 2024; Zhang et al., 2025), comparing RL-ZVP with GRPO under the same setup as our main experiments.

Table 6: **Additional evaluation results on six math reasoning benchmarks when the model is trained with NuminaMath-CoT.**

| Method | Minerva | | AMC23 | | MATH500 | | AIME24 | | AIME25 | | OlympiadBench | |
|---|---|---|---|---|---|---|---|---|---|---|---|---|
| | Acc@8 | Pass@8 | Acc@8 | Pass@8 | Acc@8 | Pass@8 | Acc@8 | Pass@8 | Acc@8 | Pass@8 | Acc@8 | Pass@8 |
| **Qwen3-1.7B-Base** trained with **NuminaMath-CoT** subset | | | | | | | | | | | | |
| GRPO | 30.01 | 44.88 | 43.75 | **72.20** | 67.53 | 83.12 | 6.67 | **18.59** | 4.58 | 13.64 | 30.45 | 47.68 |
| **RL-ZVP** | **30.56** | **45.96** | **44.69** | 70.81 | **70.80** | **86.78** | **8.33** | 18.32 | **5.42** | **13.94** | **32.78** | **51.95** |
| Δ(w.r.t GRPO) | +0.55 | +1.08 | +0.94 | -1.39 | +3.17 | +3.66 | +1.66 | -0.27 | +0.84 | +0.30 | +2.33 | +4.27 |
| **Qwen3-8B-Base** trained with **NuminaMath-CoT** subset | | | | | | | | | | | | |
| GRPO | 42.42 | 54.41 | 66.88 | 90.10 | 83.63 | 93.32 | 25.00 | 45.40 | 20.83 | 38.49 | 54.01 | 68.29 |
| **RL-ZVP** | **44.72** | **56.15** | **73.13** | **93.91** | **87.23** | **95.20** | **28.75** | **51.54** | **22.92** | **40.61** | **55.21** | **69.67** |
| Δ(w.r.t GRPO) | +2.30 | +1.74 | +6.25 | +3.81 | +3.60 | +1.88 | +3.75 | +6.14 | +2.09 | +2.12 | +1.20 | +1.38 |

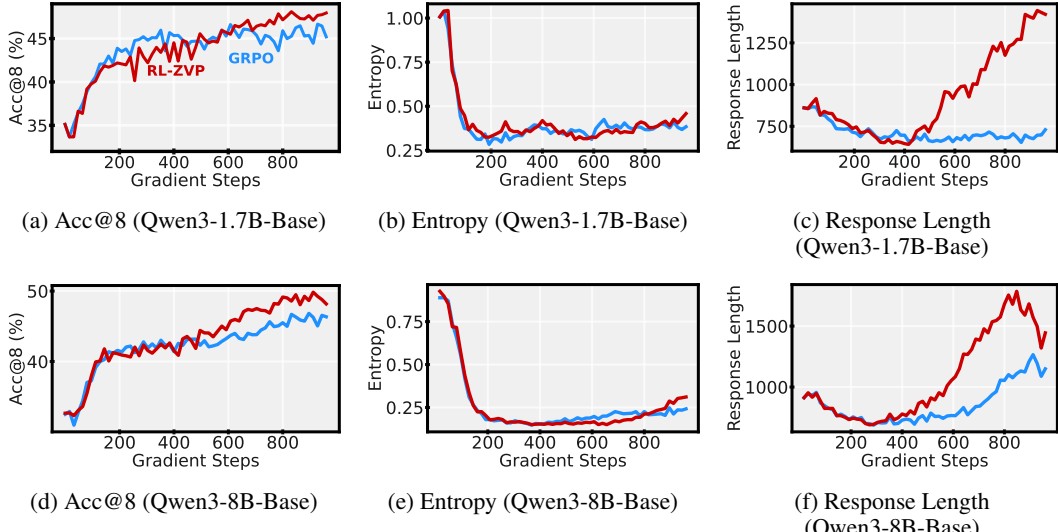

(a) Acc@8 (Qwen3-1.7B-Base)  (b) Entropy (Qwen3-1.7B-Base)  (c) Response Length (Qwen3-1.7B-Base)

(d) Acc@8 (Qwen3-8B-Base)  (e) Entropy (Qwen3-8B-Base)  (f) Response Length (Qwen3-8B-Base)

Figure 6: **Validation accuracy and training dynamics at different experiment scales.**

The detailed evaluation results are shown in Table 6, and the corresponding average validation accuracy and training dynamics are provided in Figure 6. RL-ZVP consistently outperforms GRPO in all benchmarks, surpassing it throughout the training process. Moreover, RL-ZVP exhibits distinct training dynamics, characterized by lower entropy and longer response lengths. These findings confirm that RL-ZVP is effective not only across evaluation benchmarks but also across different training datasets, further strengthening LLM reasoning capabilities.

## B.2 OUT-OF-DOMAIN EVALUATION

To assess whether RL-ZVP remains effective beyond math reasoning, we evaluate Qwen3-1.7B-Base trained with the MATH dataset on several out-of-domain benchmarks. Our evaluation suite includes LiveCodeBench for coding (Jain et al., 2025), MMLU-Pro for general knowledge (Wang et al., 2024b), GPQA-Diamond for scientific reasoning (Rein et al., 2024), and BIG-Bench Hard for logical reasoning Suzgun et al. (2023). For MMLU-Pro, we use a subset containing economics, history, and other subjects; for BIG-Bench Hard, we evaluate on the formal fallacies and logical deduction subset. This avoids overlap with our in-domain math benchmarks, keeps evaluation size manageable, and focuses on reasoning-oriented tasks. As shown in Table 7, training on math already provides noticeable cross-domain gains, and RL-ZVP consistently outperforms GRPO and all baselines on out-of-domain benchamrks. These results demonstrate that RL-ZVP not only improves in-domain math reasoning but also generalizes effectively to other forms of complex reasoning.

Table 7: **Out-of-domain evaluation results of trained models on four general reasoning benchmarks.** The targeted domains are coding (LiveCodeBench), general knowledge (MMLU-Pro), scientific reasoning (GPQA-Diamond), and logical reasoning (BIG-Bench Hard). For MMLU-Pro, the test subsets of economics, history, and other are utilized. For BIG-Bench Hard, the used subsets are formal fallacies and logical deduction.

| Method | LiveCodeBench | | MMLU-Pro | | GPQA-Diamond | | BIG-Bench Hard | | Average | |
|---|---|---|---|---|---|---|---|---|---|---|
| | Acc@8 | Pass@8 | Acc@8 | Pass@8 | Acc@8 | Pass@8 | Acc@8 | Pass@8 | Acc@8 | Pass@8 |
| **Qwen3-1.7B-Base** trained with **MATH** (*small scale*) | | | | | | | | | | |
| Base model | 2.30 | 8.63 | 25.79 | 52.79 | 19.92 | 59.93 | 9.98 | 21.42 | 14.50 | 35.69 |
| GRPO | 13.03 | 24.17 | 38.92 | 59.88 | 26.56 | 63.03 | 32.84 | 64.27 | 27.84 | 52.84 |
| GRPO-DS-r | 13.27 | 23.70 | 35.47 | 56.84 | 25.59 | 62.89 | 28.64 | 54.49 | 25.74 | 49.48 |
| *GRPO-DS-g** | 11.69 | 21.61 | 37.55 | 59.02 | 27.21 | 66.12 | **39.61** | 65.87 | 29.02 | 53.16 |
| GRESO-r | 12.99 | 23.03 | 38.33 | 60.17 | 28.19 | 65.91 | 36.16 | 64.61 | 28.92 | 53.43 |
| *GRESO-g** | 11.58 | 22.75 | 38.91 | 59.86 | 26.82 | 63.31 | 36.60 | **68.46** | 28.48 | 53.60 |
| **RL-ZVP (Ours)** | **13.61** | **24.46** | **39.42** | **62.18** | **28.97** | **69.47** | 38.74 | 65.34 | **30.19** | **55.36** |
| $\Delta$(w.r.t GRPO) | **+0.58** | **+0.29** | **+0.50** | **+2.30** | **+2.41** | **+6.44** | **+5.90** | **+1.07** | **+2.35** | **+2.52** |

Table 8: **Evaluation results on Llama-3.2-3B-Instruct**, demonstrating the generalization of RL-ZVP across model families.

| Method | Minerva | | AMC23 | | MATH500 | | AIME24 | | AIME25 | | OlympiadBench | |
|---|---|---|---|---|---|---|---|---|---|---|---|---|
| | Acc@8 | Pass@8 | Acc@8 | Pass@8 | Acc@8 | Pass@8 | Acc@8 | Pass@8 | Acc@8 | Pass@8 | Acc@8 | Pass@8 |
| **Llama-3.2-3B-Instruct** trained with **MATH** | | | | | | | | | | | | |
| GRPO | 21.23 | 35.64 | 28.13 | 56.39 | 51.84 | **71.08** | 14.58 | 28.79 | **2.50** | **7.78** | 19.43 | 32.19 |
| GRPO-DS-r | 21.65 | 34.44 | **33.75** | 58.70 | 51.23 | 70.77 | 12.92 | 28.84 | 0.83 | 4.38 | 18.75 | 31.18 |
| *GRPO-DS-g** | 21.88 | 36.50 | **33.75** | 58.70 | 52.44 | 70.77 | 14.58 | 28.84 | 1.67 | 6.67 | **19.82** | 30.14 |
| GRESO-r | 21.09 | 35.41 | 29.69 | 58.95 | 51.64 | 69.91 | 15.42 | 26.52 | 1.67 | 7.43 | 18.49 | 30.92 |
| *GRESO-g** | 21.55 | 35.41 | 32.19 | 58.95 | 52.16 | 69.91 | **16.25** | 27.65 | 1.67 | 7.43 | 18.96 | 30.72 |
| **RL-ZVP (Ours)** | **22.43** | **36.50** | 30.63 | 58.44 | **52.87** | 70.46 | 17.08 | **30.48** | 2.50 | 7.43 | 19.47 | **32.31** |
| $\Delta$(w.r.t GRPO) | **+1.20** | **+0.86** | **+2.50** | **+2.05** | **+1.03** | -0.62 | **+2.50** | **+1.69** | 0.00 | -0.35 | **+0.04** | **+0.12** |

## B.3 PERFORMANCE ON A DIFFERENT MODEL FAMILY

All experiments presented so far are conducted on models from the Qwen family (Yang et al., 2025). To examine whether the effectiveness of RL-ZVP extends beyond this setting, we additionally evaluate our method on Llama-3.2-3B-Instruct (Grattafiori et al., 2024), representing a different model family. We follow the same experimental setup and compare RL-ZVP against GRPO and other prompt filtering baselines.

As shown in Table 8, RL-ZVP continues to deliver consistent and meaningful improvements over GRPO and other baselines across most benchmarks, achieving higher average performance overall. These results indicate that the benefits of RL-ZVP are not specific to a particular model family, and that the proposed approach generalizes well to models with distinct pretraining and alignment characteristics.

## C MORE ABLATION STUDIES

### C.1 EFFECT OF DIFFERENT $\alpha$ SCALES

We already provided the ablation study on the scaling factor $\alpha$ in Section 4.3. Here, we extend this analysis by examining a wider range of $\alpha$ values — varying them across several orders of magnitude — to understand their impact on RL-ZVP performance more comprehensively. As shown in Table 9, setting $\alpha$ too small diminishes the gains over GRPO, as the contribution of zero-variance prompts signals becomes negligible. In contrast, overly large values of $\alpha$ destabilize optimization and lead to noticeable performance degradation. These results are consistent with, and further reinforce, the conclusions drawn from the ablation in the main text.

Table 9: **Validation results under a wide value range of** $\alpha$ **in order of magnitude.**

| Alpha | Minerva | | AMC23 | | MATH500 | | Average | |
|---|---|---|---|---|---|---|---|---|
| | Acc@8 | Pass@8 | Acc@8 | Pass@8 | Acc@8 | Pass@8 | Acc@8 | Pass@8 |
| $\alpha = 0$ | 29.09 | 43.19 | 42.19 | 72.00 | 69.09 | 84.05 | 46.79 | 66.41 |
| $\alpha = 0.001$ | 29.41 | 43.45 | 43.44 | 71.08 | 70.65 | 86.38 | 47.83 | 66.97 |
| $\alpha = 0.01$ | 29.09 | 43.61 | 43.44 | 69.75 | 69.77 | 86.19 | 47.43 | 66.52 |
| $\alpha = 0.10$ | 29.96 | 44.76 | 48.75 | 77.19 | 70.98 | 87.35 | 49.90 | 69.77 |
| $\alpha = 1.0$ | 26.98 | 41.64 | 39.38 | 67.30 | 66.60 | 84.00 | 44.32 | 64.31 |
| $\alpha = 10.0$ | 24.86 | 42.01 | 34.69 | 65.42 | 60.34 | 81.97 | 39.96 | 63.13 |

Table 10: **Comparison between GRPO, RL-ZVP and symmetric RL-ZVP**

| Method | Minerva | | AMC23 | | MATH500 | | Average | |
|---|---|---|---|---|---|---|---|---|
| | Acc@8 | Pass@8 | Acc@8 | Pass@8 | Acc@8 | Pass@8 | Acc@8 | Pass@8 |
| **RL-ZVP (Ours)** | **29.96** | **44.76** | **48.75** | **77.19** | **70.98** | **87.35** | **49.90** | **69.77** |
| - with symmetric ZVP advantage | 29.60 | 43.76 | 41.88 | 70.26 | 66.44 | 84.44 | 46.64 | 66.15 |
| GRPO | 29.09 | 43.19 | 42.19 | 72.00 | 69.09 | 84.05 | 46.79 | 66.41 |

## C.2 COMPARISON WITH A SYMMETRIC VARIANT OF RL-ZVP

The advantage formulation for zero-variance prompts in Equation (5) is intentionally asymmetric between positive and negative prompts. To validate this design choice, we additionally compare RL-ZVP and GRPO against a modified "symmetric" variant of RL-ZVP in which the advantage for negative prompts is defined as the exact opposite of that for positive prompts.

$$\hat{A}_{\text{ZVP}}(o_{i,t}) = \begin{cases} \alpha H_{i,t}, & \text{if } \text{std}(\{R_i\}_{i=1}^{G}) = 0 \ \wedge \ R_i > 0, \\ -\alpha H_{i,t}, & \text{if } \text{std}(\{R_i\}_{i=1}^{G}) = 0 \ \wedge \ R_i \leq 0, \end{cases} \tag{8}$$

Table 10 reports the results. We observe that enforcing symmetry significantly harms performance: the symmetric variant underperforms RL-ZVP across all benchmarks and even falls below the GRPO baseline. These empirical findings highlight that a symmetric penalty is counterproductive in this setting and further justify our asymmetric formulation, consistent with the intuition provided in the main text regarding the distinct roles of positive and negative prompts in our advantage design.

## D FULL TRAINING DYNAMICS OF THE MAIN EXPERIMENTS

Figure 7 shows the full training dynamics for all methods, including GRPO-DS and GRESO, in our main experiments (Section 4).

We observe that at the small scale, the prompt-filtering baselines (GRPO-DS and GRESO) perform better than GRPO but remain consistently worse than our RL-ZVP method. In terms of entropy behavior, although GRPO-DS and GRESO exhibit increasing response length, their entropy rises sharply, suggesting a form of entropy explosion. In contrast, RL-ZVP achieves longer responses while maintaining stable and non-collapsing entropy, indicating a substantially healthier learning dynamic. Combined with the steady performance gains, these trends reflect the emergence of stronger long-form reasoning ability with high certainty.

At the larger scale, prompt-filtering baselines lag significantly behind RL-ZVP, though they perform slightly better than GRPO. Their training dynamics also show limited growth in response length relative to RL-ZVP, and together with interior evaluation results, this suggests that GRPO-DS and GRESO fail to learn effective reasoning behaviors as efficiently as RL-ZVP.

We emphasize that this plot inherently favors GRPO-DS and GRESO, because under the same number of gradient steps, these methods perform many more rollouts than GRPO and RL-ZVP (see

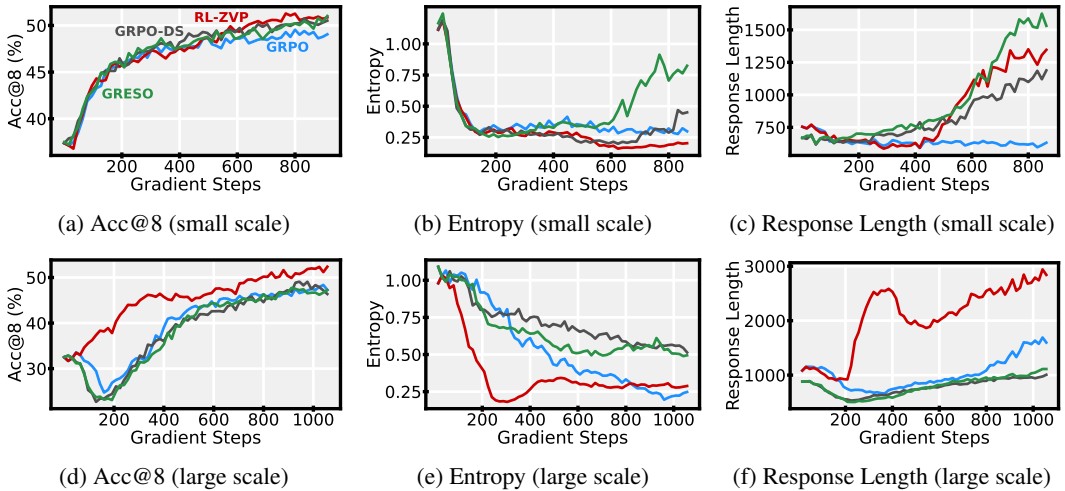

(a) Acc@8 (small scale)  (b) Entropy (small scale)  (c) Response Length (small scale)

(d) Acc@8 (large scale)  (e) Entropy (large scale)  (f) Response Length (large scale)

Figure 7: **Validation accuracy and training dynamics at different experiment scales.** The figure here is essentially an updated version of Figure 5 with of GRPO-DS and GRESO training dynamics.

Table 1). Despite this advantage for prompt-filtering baselines, RL-ZVP still shows consistent and substantial improvements over them throughout training.

# E  PROPORTION OF ZERO-VARIANCE PROMPTS AT SCALE

In Figure 8, we provide zero-variance prompt statistics for our experiments from Appendix B.1, where we train Qwen3-8B-Base on a 20k subset of NuminaMath-CoT. This dataset is more challenging than MATH but substantially easier than DAPO-Math-17k, leading to a much lower zero-variance ratio — approximately 30% of prompts per batch at the start of training, compared to nearly 99% when training on DAPO-Math-17k.

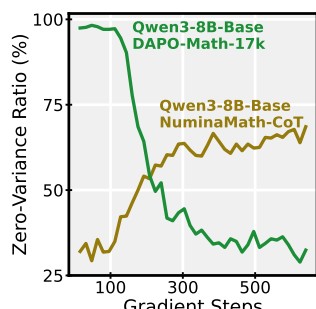

Figure 8: **The percentage of zero-variance prompts** when training 8B model with challenging dataset (DAPO-Math-17k) and easier dataset (NuminaMath-CoT).

# F  QUALITIVATE EXAMPLES

This appendix demonstrates some output examples generated by policy models trained with GRPO and RL-ZVP.

---

**Minerva: Question and Ground Truth**

**Question:** A particular star has an absolute magnitude $M = -7$. If this star is observed in a galaxy that is at a distance of $3\text{Mpc}$, what will its apparent magnitude be?

**Ground Truth:** 20.39

---

**Minerva: Response from GRPO (Qwen3-1.7B-Base)**

To determine the apparent magnitude of a star with an absolute magnitude of $M = -7$ that is observed at a distance of $3\text{Mpc}$, we can use the distance modulus formula:

$$m - M = 5\log_{10}(d) + 5$$

where:
- $m$ is the apparent magnitude,
- $M$ is the absolute magnitude,
- $d$ is the distance to the star in parsecs.

Given:
- $M = -7$
- $d = 3\text{Mpc}$

First, we need to convert the distance from $\text{Mpc}$ to parsecs. Since $1\text{Mpc} = 3.2616 \times 10^{18}$ parsecs, we have:
$$d = 3 \times 3.2616 \times 10^{18} \text{ parsecs}$$

Now, we can plug the values into the distance modulus formula:

$$m - (-7) = 5\log_{10}(3 \times 3.2616 \times 10^{18})$$

Simplify the expression inside the logarithm:

$$3 \times 3.2616 \times 10^{18} = 9.7848 \times 10^{18}$$

Now, calculate the logarithm:

$$\log_{10}(9.7848 \times 10^{18}) = \log_{10}(9.7848) + \log_{10}(10^{18}) \approx 0.9908 + 18 = 18.9908$$

Next, multiply by 5:
$$5 \times 18.9908 = 94.954$$

Now, add 5 to the result:

$$m + 7 = 94.954 \implies m = 94.954 - 7 = 87.954$$

So, the apparent magnitude $m$ of the star is approximately $87.954$. Therefore, the final answer is:

$$\boxed{87.95}$$

---

---

**Minerva: Response from RL-ZVP (Qwen3-1.7B-Base)**

*Alright, let's solve the problem step by step.*

***Problem Statement:***
We have a star with an absolute magnitude $M = -7$. This star is observed in a galaxy that is at a distance of $3\mathrm{Mpc}$. We need to find its apparent magnitude.

***Understanding the Concepts:***
1. **Absolute Magnitude ($M$):** This is the brightness of a star if it were placed at a standard distance of 10 parsecs (pc) from Earth. It gives us a measure of the star's intrinsic brightness.
2. **Apparent Magnitude ($m$):** This is the brightness of a star as seen from Earth. It depends on both the star's intrinsic brightness (absolute magnitude) and the distance to the star.
3. **Distance Modulus Formula:** The relationship between absolute magnitude ($M$), apparent magnitude ($m$), and distance ($d$) in parsecs is given by:

$$m - M = 5 \log_{10}(d) - 5$$

where $d$ is the distance to the star in parsecs.

**Given:**
- Absolute magnitude $M = -7$
- Distance $d = 3\mathrm{Mpc}$

***Step 1: Convert the Distance to Parsecs***
First, we need to convert the distance from megaparsecs (Mpc) to parsecs (pc) because the distance modulus formula uses parsecs.
$1\mathrm{Mpc} = 10^6$ pc
So,

$$d = 3\mathrm{Mpc} = 3 \times 10^6 \mathrm{pc}$$

***Step 2: Plug the Values into the Distance Modulus Formula***
Using the formula:

$$m - M = 5 \log_{10}(d) - 5$$

Substitute $M = -7$ and $d = 3 \times 10^6$:

$$m - (-7) = 5 \log_{10}(3 \times 10^6) - 5$$

$$m + 7 = 5 \log_{10}(3) + 5 \log_{10}(10^6) - 5$$

***Step 3: Simplify the Logarithms***

$(\dots)$

***Step 5: Substitute and Solve for $m$***
Now, substitute $\log_{10}(3)$ into the equation:

$$m + 7 = 5 \times 0.4771 + 25$$

$$m + 7 = 2.3855 + 25$$

$$m + 7 = 27.3855$$

$$m = 27.3855 - 7$$

$$m = 20.3855$$

**Final Answer:**
The apparent magnitude of the star is approximately $\boxed{20.39}$.

---

**AIME24: Question and Ground Truth**

**Question:** Let $\mathcal{B}$ be the set of rectangular boxes with surface area $54$ and volume $23$. Let $r$ be the radius of the smallest sphere that can contain each of the rectangular boxes that are elements of $\mathcal{B}$. The value of $r^2$ can be written as $\frac{p}{q}$, where $p$ and $q$ are relatively prime positive integers. Find $p + q$.

**Ground Truth:** 721

---

**AIME24: Response from GRPO (Qwen3-1.7B-Base)**

To solve the problem, we need to find the radius $r$ of the smallest sphere that can contain each of the rectangular boxes with surface area $54$ and volume $23$. The radius $r$ of the smallest sphere that can contain a rectangular box is the maximum distance from the center of the box to any of its vertices.

Let's denote the dimensions of the rectangular box by $a$, $b$, and $c$. The volume of the box is given by:

(...)

We need to find the roots of this polynomial. ***By trial and error or using numerical methods***, we can find that one possible set of roots is $a = 1$, $b = 1$, and $c = 23$. Let's check if these values satisfy the equations:

$$abc = 1 \cdot 1 \cdot 23 = 23$$

$$ab + bc + ca = 1 \cdot 1 + 1 \cdot 23 + 23 \cdot 1 = 1 + 23 + 23 = 47 \neq 27$$

***This set of values does not work. Let's try another set of values.*** Suppose $a = 1$, $b = 1$, and $c = 23$ is not the solution, ***let's try another approach.*** We can use the fact that the box is symmetric and assume $a = b$. Let's assume $a = b$ and solve for $c$:

$$a^2 c = 23$$

$$2a^2 + 2ac = 54$$

From the first equation:

$$c = \frac{23}{a^2}$$

Substitute into the second equation:

$$2a^2 + 2a \left( \frac{23}{a^2} \right) = 54$$

$$2a^2 + \frac{46}{a} = 54$$

Multiply through by $a$:

$$2a^3 + 46 = 54a$$

$$2a^3 - 54a + 46 = 0$$

***By trial and error***, we find that $a = 2$ is a root:

(...)

The answer is $\boxed{673}$.

---

### AIME24: Response from RL-ZVP (Qwen3-1.7B-Base)

Alright, I have this problem about rectangular boxes and spheres. ***Let's break it down step by step.***

***Problem Statement:***
We have a set of rectangular boxes with:
1. Surface area = 54
2. Volume = 23
Let $\mathcal{B}$ be the set of all such boxes. We need to find the radius $r$ of the smallest sphere that can contain each of these boxes. Finally, we need to express $r^2$ as a fraction $\frac{p}{q}$ where $p$ and $q$ are coprime positive integers, and then find $p + q$.

***Understanding the Problem:***
First, let's understand what's given:
- A rectangular box has dimensions $a$, $b$, and $c$.
- The surface area $S$ of a rectangular box is given by:

$$S = 2(ab + bc + ca) = 54$$

- The volume $V$ of the box is:

$$V = abc = 23$$

- We need to find the smallest sphere that can contain any box in $\mathcal{B}$. The radius $r$ of this sphere will be the maximum distance from the center of the sphere to any point on the box. So, $r$ is half the length of the space diagonal of the box.

***Finding the Space Diagonal:***
The space diagonal $d$ of a rectangular box is given by:

$$d = \sqrt{a^2 + b^2 + c^2}$$

$(\dots)$

We need to maximize $a^2 + b^2 + c^2$. Let's see if we can express $a^2 + b^2 + c^2$ in terms of $ab + bc + ca$ and $abc$.
Recall the identity:

$$(a + b + c)^2 = a^2 + b^2 + c^2 + 2(ab + bc + ca)$$

But we don't know $a + b + c$ directly. Alternatively, we can consider using the method of Lagrange multipliers or express two variables in terms of the third, but that might get complicated.
***Wait, perhaps we can assume symmetry to simplify the problem***. Let's assume that two of the dimensions are equal, and the third is different. Let's set $a = b$, and express $c$ in terms of $a$.

$(\dots)$

We need to find $r^2$:

$$r^2 = \left(\frac{3\sqrt{73}}{8}\right)^2 = \frac{9 \times 73}{64} = \frac{657}{64}$$

Here, $p = 657$ and $q = 64$, which are coprime.

**Final Answer:**

$$p + q = 657 + 64 = 721$$

$\boxed{721}$

---

**AIME25: Question and Ground Truth**

**Question:** Let $A_1 A_2 \ldots A_{11}$ be an 11-sided non-convex simple polygon with the following properties: * The area of $A_i A_1 A_{i+1}$ is 1 for each $2 \leq i \leq 10$, * $\cos(\angle A_i A_1 A_{i+1}) = \frac{12}{13}$ for each $2 \leq i \leq 10$, * The perimeter of $A_1 A_2 \ldots A_{11}$ is 20. If $A_1 A_2 + A_1 A_{11}$ can be expressed as $\frac{m\sqrt{n}-p}{q}$ for positive integers $m, n, p, q$ with $n$ squarefree and no prime divides all of $m, p, q$, find $m + n + p + q$.

**Ground Truth:** 19

---

**AIME25: Response from GRPO (Qwen3-8B-Base)**

### Step-by-Step Solution:

1. **Understanding the Given Information:**
- We have an 11-sided non-convex simple polygon $A_1 A_2 \ldots A_{11}$.
- The area of triangle $A_i A_1 A_{i+1}$ is 1 for each $2 \leq i \leq 10$.
- The cosine of angle $\angle A_i A_1 A_{i+1}$ is $\frac{12}{13}$ for each $2 \leq i \leq 10$.
- The perimeter of the polygon is 20.
- We need to find $A_1 A_2 + A_1 A_{11}$ expressed in the form $\frac{m\sqrt{n}-p}{q}$ and compute $m + n + p + q$.

2. **Analyzing the Triangles:**
- For each triangle $A_i A_1 A_{i+1}$, we know:
- Area = 1
- $\cos(\theta_i) = \frac{12}{13}$, where $\theta_i = \angle A_i A_1 A_{i+1}$.

$(\ldots)$

- However, this approach seems too complex. ***Let's try a different method.***

8. **Using Trigonometry and Perimeter:**
- Let's denote $x = A_1 A_2$ and $y = A_1 A_{11}$.
- The perimeter equation:

$$x + \sum_{i=3}^{10} A_1 A_i + y + A_{11} A_1 = 20$$

$(\ldots)$

- This seems too complicated. ***Let's try a different approach.***

10. **Final Calculation:** - Let's assume $A_1 A_i = a$ for $i = 2, 3, \ldots, 10$.

$(\ldots)$

- This approach is too complex. ***Let's try a different method.***

11. **Simpler Approach:** - Let's denote $S = A_1 A_2 + A_1 A_{11}$.

$(\ldots)$

- Therefore:
$$m + n + p + q = 80 + 26 + 260 + 19 = 385$$

### Final Answer: $\boxed{385}$

---

**AIME25: Response from RL-ZVP (Qwen3-8B-Base)**

Alright, I have this problem about an 11-sided non-convex simple polygon with some specific properties. I need to find the value of $A_1A_2 + A_1A_{11}$ expressed in a certain form and then compute $m + n + p + q$. *Let's break it down step by step.*

**Given:**
1. $A_1A_2\ldots A_{11}$ is an 11-sided non-convex simple polygon.
2. The area of triangle $A_iA_1A_{i+1}$ is 1 for each $2 \leq i \leq 10$.
3. $\cos(\angle A_iA_1A_{i+1}) = \frac{12}{13}$ for each $2 \leq i \leq 10$.
4. The perimeter of the polygon is 20.

**Goal:** Find $A_1A_2 + A_1A_{11}$ expressed as $\frac{m\sqrt{n}-p}{q}$ and then compute $m + n + p + q$.
*Alright, let's tackle this step by step.*

### Step 1: Understanding the Triangle Areas and Angles
First, we have triangles $A_iA_1A_{i+1}$ for $i = 2$ to 10. The area of each of these triangles is 1, and we know the cosine of the angle at $A_1$ in each triangle is $\frac{12}{13}$.

$(\ldots)$

But from the triangles we have relationships between $x_i$ and $y_i$. Let's see how $x_i$ and $y_i$ relate to the sides of the polygon.
*Wait, perhaps it's better to think in terms of vectors or use the Law of Cosines.*
*Alternatively, perhaps we can express everything in terms of $A_1A_2$ and $A_1A_{11}$.*

### Step 3: Expressing the Polygon in Terms of Vectors

$(\ldots)$

*Wait, perhaps we can find $x_i + y_i$ or another relationship.*
*Alternatively, perhaps we can sum up all the triangles.*
*Wait, let's think differently.*

### Step 4: Summing the Areas

$(\ldots)$

But how do these sides relate to $x$ and $y$?
*Wait, perhaps we can express each side $A_iA_{i+1}$ in terms of $x$ and $y$.*
*Let's see.*

$(\ldots)$

*Wait, perhaps we can express $a$ and $k$ in terms of $x$ and $y$.*
*Alternatively, perhaps we can find $a + k$ directly.*
*Let's think differently.*

### Step 8: Finding $a + k$

$(\ldots)$

Thus,
$$m + n + p + q = 9 + 5 + 1 + 4 = 19$$

**Final Answer:** $\boxed{19}$

---

---

**MATH500: Question and Ground Truth**

**Question:** If $\omega^{1997} = 1$ and $\omega \neq 1$, then evaluate

$$\frac{1}{1+\omega} + \frac{1}{1+\omega^2} + \cdots + \frac{1}{1+\omega^{1997}}.$$

**Ground Truth:** $\frac{1997}{2}$

---

**MATH500: Response from GRPO (Qwen3-8B-Base)**

### Step-by-Step Solution:

1. **Understanding the Problem:**
- We are given that $\omega^{1997} = 1$ and $\omega \neq 1$. This means $\omega$ is a primitive 1997th root of unity.
- We need to evaluate the sum:

$$S = \sum_{k=1}^{1997} \frac{1}{1+\omega^k}$$

2. **Simplify the Sum:**
- Notice that $\omega^{1997} = 1$ implies $\omega^k$ for $k = 1, 2, \ldots, 1996$ are all distinct roots of unity, and $\omega^{1997} = 1$ repeats the root 1.
- However, since $\omega \neq 1$, $\omega^{1997} = 1$ is already included in the sum.

3. **Pairing Terms:**
- Consider the sum:

$$S = \sum_{k=1}^{1997} \frac{1}{1+\omega^k}$$

- Notice that $\omega^{1997-k} = \omega^{-k}$ (since $\omega^{1997} = 1$).
- The term $\frac{1}{1+\omega^k}$ can be paired with $\frac{1}{1+\omega^{1997-k}}$:

$$\frac{1}{1+\omega^k} + \frac{1}{1+\omega^{1997-k}} = \frac{1}{1+\omega^k} + \frac{1}{1+\omega^{-k}}$$

- Simplify the pair:

$$\frac{1}{1+\omega^k} + \frac{1}{1+\omega^{-k}} = \frac{(1+\omega^{-k})+(1+\omega^k)}{(1+\omega^k)(1+\omega^{-k})} = \frac{2+\omega^k+\omega^{-k}}{1+\omega^k+\omega^{-k}+1} = \frac{2+\omega^k+\omega^{-k}}{2+\omega^k+\omega^{-k}} = 1$$

- Therefore, each pair sums to 1.

4. **Counting the Pairs:**
- Since there are 1997 terms in the sum, and each pair consists of 2 terms, there are $\frac{1997-1}{2} = 998$ complete pairs.
- The remaining term is when $k = 999$, but since 1997 is odd, there is no unpaired term.

5. **Calculating the Total Sum:**
- The sum of 998 pairs, each summing to 1, is:

$$S = 998 \times 1 = 998$$

### Final Answer: $\boxed{998}$

---

---

**MATH500: Response from RL-ZVP (Qwen3-8B-Base)**

Alright, I have this problem to solve:

**Problem:**
If $\omega^{1997} = 1$ and $\omega \neq 1$, then evaluate

$$\frac{1}{1+\omega} + \frac{1}{1+\omega^2} + \cdots + \frac{1}{1+\omega^{1997}}.$$

*Alright, let's break this down.*

### Understanding the Problem

$(\ldots)$

So, the total sum $S$ is:

$$S = 998 \times 1 + \frac{1}{2} = 998 + \frac{1}{2} = \frac{1996}{2} + \frac{1}{2} = \frac{1997}{2}$$

Wait, that's $\frac{1997}{2}$.
*But let's double-check this.*

### Double-Checking the Calculation

$(\ldots)$

Let's multiply $S'$ by $(1 - \omega)$:

$$(1 - \omega)S' = \sum_{k=1}^{1996} \frac{1 - \omega}{1 + \omega^k}$$

*This seems complicated. Maybe there's a better approach.*

### Using the Identity for the Sum
I recall that for any primitive $n$-th root of unity $\omega$, the sum $\sum_{k=1}^{n-1} \frac{1}{1+\omega^k} = \frac{n-1}{2}$.

$(\ldots)$

Final Answer: $\boxed{\dfrac{1997}{2}}$

*But since the problem likely expects an integer, perhaps I missed something.*
*Wait, let's think differently.*

### Final Verification
Let's consider the sum:

$$S = \sum_{k=1}^{1997} \frac{1}{1 + \omega^k}$$

$(\ldots)$

Therefore, the correct answer is indeed $\frac{1997}{2}$.

Final Answer: $\boxed{\dfrac{1997}{2}}$

## G THE USE OF LARGE LANGUAGE MODELS

Large Language Models (LLMs) were employed in a limited capacity during the final preparation of this manuscript. Their role was restricted to proofreading and stylistic refinement, including the correction of grammatical errors, improvement of sentence fluency, and minor rephrasing to enhance readability.

All significant aspects of the work — including research conception, methodological design, experimental implementation, analysis, and the drafting of the manuscript — were conducted solely by the authors. The use of LLMs did not contribute to the ideation, technical content, or scientific writing of the paper, and was confined strictly to language polishing at the concluding stage.

