# OpenReview forum: "No Prompt Left Behind: Exploiting Zero-Variance Prompts in LLM Reinforcement Learning via Entropy-Guided Advantage Shaping"
_ICLR.cc/2026/Conference — ICLR 2026 Poster_

### Official Review · Reviewer_3GQ5 · 2025-10-29

**Soundness:** 3
**Presentation:** 3
**Contribution:** 3
**Rating:** 6
**Confidence:** 3

**Summary:**

This paper addresses a critical inefficiency in Reinforcement Learning with Verifiable Rewards (RLVR), known as the "advantage vanishing" problem. Standard methods, such as GRPO, fail to learn from "zero-variance prompts"—instances where all sampled responses are either uniformly correct or incorrect—because their advantage calculation collapses to zero. The authors compellingly argue that this constitutes a significant source of inefficiency, as these prompts are both computationally expensive (rollouts consume approximately 50% of training time) and highly prevalent (comprising 30-99% of the data). The proposed method, RL-ZVP, leverages these prompts rather than filtering them. It achieves this by defining a new advantage signal: the direction is set to positive for all-correct groups and negative for all-incorrect groups, while the magnitude is modulated by token-level entropy. This "entropy-guided" shaping mechanism rewards high-entropy (uncertain, complex) tokens in correct responses and penalizes low-entropy (confident) tokens in incorrect responses. Rigorous experiments demonstrate that RL-ZVP significantly outperforms GRPO. Notably, it also surpasses modern filtering-based baselines (e.g., GRPO-DS, GRESO), even when those baselines are allocated 3–5 times more computational budget for rollouts.

**Strengths:**

The paper's primary strength lies in its novel and intuitive solution to a practical and well-defined problem. The entropy-guided advantage shaping mechanism is a well-motivated and insightful design that effectively rewards productive, uncertain reasoning steps while penalizing confident errors. This strong methodology is validated by a particularly rigorous experimental setup. The authors compare RL-ZVP not only to the standard GRPO but also to two strong baselines (GRPO-DS, GRESO) specifically designed to filter the problematic prompts. By testing under both equal-rollout (fair) and equal-gradient-step (baseline-favored) conditions, the paper offers compelling evidence for RL-ZVP's superior performance and efficiency. This quantitative strength is substantiated by in-depth analysis, including an examination of training dynamics that explains its enhanced stability (leveraging signals from both easy and hard prompts at different training stages) and qualitative examples (Appendix D) that demonstrate a tangible improvement in reasoning correctness, not just stylistic complexity.

**Weaknesses:**

1）The method proposed is highly sensitive to the $\alpha$ hyper-parameter. The experiments results in Table 3 demonstrate that the optimal performance window for the scaling factor $\alpha$ is very narrow. This raises concerns that the method may require extensive and precise fine-tuning when applied to different datasets or models.

2) The description on negative prompts seems confusing. The logic for advantage shaping (Section 3.2) is asymmetrical. Positive prompts reward high-entropy tokens far more than  negative prompts penalize high-entropy tokens. Could the author explain why use this setting?

3 I think it will be better to provide more sufficient ablation Studies. The ablation study Table 2, while strong, omits a key comparison. It does not test a symmetrical penalty formula for negative prompts (e.g., $\hat{A} = -\alpha H_{i,t}$, which would penalize high-entropy error tokens more to see how it would affect performance.

**Questions:**

See weakness

---

> ### Author Response · Authors · 2025-11-19
> **Official Response to Reviewer 3GQ5 (1/3)**
>
> We thank the reviewer for taking the time to evaluate our submission. We are glad that you find significant merits in our work, and we sincerely appreciate the constructive feedback and insightful questions, which will help us further strengthen the manuscript. We address the reviewer's concerns as follows:
>
> > The method proposed is highly sensitive to the $\alpha$ hyper-parameter. The experiments results in Table 3 demonstrate that the optimal performance window for the scaling factor $\alpha$ is very narrow. This raises concerns that the method may require extensive and precise fine-tuning when applied to different datasets or models.
>
> Across all of our experiments, we set the scaling factor to **$\alpha = 0.1$**. The empirical results demonstrate that this choice generalizes extremely well: RL-ZVP with this fixed value consistently and significantly **outperforms GRPO** across **two model scales** (1.7B and 8B), **three training datasets** (MATH, DAPO-Math-17k, NuminaMath-CoT), and **six evaluation benchmarks**. We believe these findings may **address the reviewer's concern about whether RL-ZVP would require extensive or precise hyperparameter tuning when applied to different models or datasets**: our results show that $\alpha = 0.1$ works robustly without the need for adjustment.
>
> Additionally, we would like to provide more context on how we originally selected the appropriate value for $\alpha$. We first conducted a preliminary GRPO training run to measure the entropy distribution of the base model's generated tokens. Empirically, we observed that the 100th percentile of token entropy typically lies in the range [9, 11], while the 80th percentile falls around [1.5, 2.5]. Based on this observation, we choose $\alpha$ such that the scaled entropy-guided advantages fall within the same magnitude as the usual GRPO-computed advantages under the same setting. We therefore selected $\alpha = 0.1$ for all experiments, and it works extremely well in practice. As a result, **no extensive hyperparameter search or tuning was required or conducted for the scaling factor**. Instead, we rely on a simple, empirically guided strategy to determine a reasonable range for $\alpha$, and in practice, we did not need to adjust this value across different experimental settings, as $\alpha = 0.1$ consistently generalizes remarkably well.
>
> Regarding the results in Table 3, as discussed in the manuscript, having a specific range for $\alpha$ is important to ensure that the learning signals from zero-variance prompts contribute effectively to RL training. This range is chosen based on empirically driven observations, rather than an intensive or brute-force hyperparameter search. Setting $\alpha$ too low diminishes the contribution of the zero-variance signal, while setting it too high leads to unstable training. Thus, having a **relatively narrow suitable range for $\alpha$ is not necessarily a drawback**; rather, it is expected and intuitive, since the purpose of $\alpha$ is to scale the zero-variance prompt advantages into a range comparable to standard GRPO-computed advantages.

---

> ### Author Response · Authors · 2025-11-19
> **Official Response to Reviewer 3GQ5 (2/3)**
>
> > I think it will be better to provide more sufficient ablation Studies. The ablation study Table 2, while strong, omits a key comparison. It does not test a symmetrical penalty formula for negative prompts (e.g., $\hat{A}=-\alpha H_{i,t}$) which would penalize high-entropy error tokens more to see how it would affect performance.
>
> We thank the reviewer for this sharp and insightful suggestion. Below, we provide the **additional experiment evaluating RL-ZVP with a symmetric penalty formulation for negative prompts**, which will be incorporated into the final version of the manuscript.
>
> $$
> {
> \\begin{array}{l|ll|ll|ll|ll|ll}
> \\hline
> \\text{Method} &
> \\text{Minerva} & &
> \\text{AMC23} & &
> \\text{MATH500} & &
> \\text{Avg} & \\\\
> &
> \\text{Acc@8} & \\text{Pass@8} &
> \\text{Acc@8} & \\text{Pass@8} &
> \\text{Acc@8} & \\text{Pass@8} &
> \\text{Acc@8} & \\text{Pass@8} \\\\
> \\hline
> \\textbf{RL-ZVP (Ours)} &
> \\mathbf{29.96} & \\mathbf{44.76} &
> \\mathbf{48.75} & \\mathbf{77.19} &
> \\mathbf{70.98} & \\mathbf{87.35} &
> \\mathbf{49.90} & \\mathbf{69.77} \\\\
> \\hline
> \\text{  w/o negative prompts} &
> 29.92 & 43.87 &
> 44.69 & 70.51 &
> 68.64 & 84.40 &
> 47.75 & 66.26 \\\\
> \\text{  w/o positive prompts} &
> 29.50 & 43.88 &
> 41.56 & 70.40 &
> 70.70 & 85.81 &
> 47.25 & 66.70 \\\\
> \\text{  w/o entropy scaling} &
> 28.95 & 43.02 &
> 42.50 & 70.76 &
> 69.19 & 84.37 &
> 46.88 & 66.05 \\\\
> \\text{  with symmetrical ZVP advantage} &
> 29.60 & 43.76 &
> 41.88 & 70.26 &
> 66.44 & 84.44 &
> 46.64 & 66.15 \\\\
> \\hline
> \\text{GRPO} &
> 29.09 & 43.19 &
> 42.19 & 72.00 &
> 69.09 & 84.05 &
> 46.79 & 66.41 \\\\
> \\hline
> \\end{array}
> }
> $$
>
>
> From the results, we observe that applying a symmetric penalty for negative prompts substantially degrades performance: RL-ZVP performs worse on all benchmarks and in fact becomes inferior to the GRPO baseline. These empirical findings further validate our choice of an asymmetric design for handling zero-variance prompts in RL-ZVP, and strongly support the intuition we provide in the subsequent explanation regarding why symmetric advantage formulation for zero-variance prompts is ineffective in this context.

---

> ### Author Response · Authors · 2025-11-19
> **Official Response to Reviewer 3GQ5 (3/3)**
>
> > The description on negative prompts seems confusing. The logic for advantage shaping (Section 3.2) is asymmetrical. Positive prompts reward high-entropy tokens far more than negative prompts penalize high-entropy tokens. Could the author explain why use this setting?
>
> Empirically, as shown above, the symmetric version of RL-ZVP performs worse than our proposed asymmetric formulation. For the intuition behind this design choice, we refer the reviewer to **Section 3.2, lines 193-199**, where we discuss the **rationale for the magnitude of the zero-variance prompt advantages**.
>
> To reiterate, high-entropy tokens (typically connectors, branching points, or other reasoning-critical tokens) play an essential role in eliciting long Chain-of-Thought behaviors such as reflection and verification. Consequently, during rewarding, we deliberately prioritize high-entropy tokens by assigning them larger advantages, thereby encouraging the model to explore and extend its reasoning process.
>
> For negative prompts, however, a symmetric formulation would penalize high-entropy tokens more severely. This would undesirably suppress the very tokens that are most important for reasoning, significantly reducing their likelihood of appearing in future reasoning traces, either in subsequent training steps or at inference time. While the algorithm should indeed decrease the log-probability of all tokens in an incorrect response in negative prompts, we do not want to over-suppress reasoning-critical, high-entropy tokens. Instead, the asymmetric design penalizes these tokens more lightly, preserving the model's ability to revisit and explore these reasoning paths in subsequent steps.
>
> In summary, the asymmetric formulation for negative-prompt advantages serves to mitigate excessive penalties on high-entropy reasoning tokens, preserving flexibility during both training and inference. This preservation is crucial for maintaining robust exploration and enabling the emergence of long-form reasoning behaviors.

---

> ### Author Response · Authors · 2025-11-26
> **A Kind Reminder**
>
> Dear reviewer 3GQ5,
>
> We would like to express our sincerest gratitude for your valuable feedback, which has greatly helped us improve our work.
>
> As the rebuttal phase is coming to an end and it has been a while since our earlier responses, we would like to follow up to see whether you have any remaining concerns or questions regarding our submission. We are more than happy to continue the discussion and clarify anything needed to further strengthen the paper.
>
> Thank you once again, and we hope you have a wonderful day.

---

### Official Review · Reviewer_xC6w · 2025-11-01

**Soundness:** 3
**Presentation:** 3
**Contribution:** 3
**Rating:** 6
**Confidence:** 3

**Summary:**

The paper aims to leverage Zero-Variance Prompts (consisting of fully correct or fully incorrect responses) during RL to enable more efficient learning. The authors propose using token-level entropy-based advantage updates for these responses. Extensive experiments are conducted to demonstrate the effectiveness of the method on reasoning tasks.

**Strengths:**

- The paper is well written and easy to follow.

- The proposed method of assigning advantages to tokens in responses to zero-variance prompts is intuitive and well motivated.

- Extensive experiments are conducted to demonstrate the effectiveness of the method on reasoning tasks.

**Weaknesses:**

- Significance of the problem: Are there any empirical statistics on the proportion of zero-variance prompts? I notice that the authors only train 8B models on the harder DAPO-Math-17k dataset. It would be helpful to also show empirical results on Math with 8B models (scenarios with fewer zero-variance prompts) to evaluate the effect of the newly designed advantage. We might expect that it does not hurt performance in scenarios with fewer zero-variance prompts, while providing greater benefits when the number of zero-variance prompts is larger.

- How is it ensured that the advantages assigned to tokens in zero-variance prompts are on the same scale as normal advantages? Could differences in scale affect training stability?

- How should an appropriate value of α be chosen? Are there any guidelines or heuristics for selecting it?

**Questions:**

Please refer to the weakness part.

---

> ### Author Response · Authors · 2025-11-19
> **Official Response to Reviewer xC6w (1/4)**
>
> We thank the reviewer for taking the time to evaluate our submission. We are glad that you find significant merits in our work, and we sincerely appreciate the constructive feedback and insightful questions, which will help us further strengthen the manuscript. We address the reviewer's concerns as follows:
>
> > Significance of the problem: Are there any empirical statistics on the proportion of zero-variance prompts? I notice that the authors only train 8B models on the harder DAPO-Math-17k dataset. It would be helpful to also show empirical results on Math with 8B models (scenarios with fewer zero-variance prompts) to evaluate the effect of the newly designed advantage. We might expect that it does not hurt performance in scenarios with fewer zero-variance prompts, while providing greater benefits when the number of zero-variance prompts is larger.
>
> This is a very sharp and insightful observation!
>
> Firstly, we would like to refer the reviewer to **Figure 3**, which directly addresses the question regarding the empirical statistics of zero-variance prompts. As shown in the figure, in the large-scale experiments using Qwen3-8B-Base on DAPO-Math-17k, nearly 99% of prompts in each batch are zero-variance at the beginning of training (when the model is not yet capable of solving the difficult problems). This proportion gradually decreases to about 35%, after which it remains stable through the rest of training.
>
> *(To be continued in next response)*

---

> ### Author Response · Authors · 2025-11-19
> **Official Response to Reviewer xC6w (2/4)**
>
> *(Continued from previous response)*
>
>
> Secondly, as mentioned in the manuscript, we consider the MATH dataset too trivial for Qwen3-8B-Base (given that it is already not challenging for Qwen3-1.7B-Base). This is why we chose DAPO-Math-17k for the main experiment. Nevertheless, the reviewer's point about evaluating RL-ZVP in settings with fewer zero-variance prompts is entirely valid and insightful. In fact, when using MATH for an 8B model, we anticipate a large proportion of positive zero-variance prompts - since the base model can already answer many questions correctly - making it unsuitable for analyzing the scenario the reviewer is interested in.
>
> To provide a meaningful setting with fewer zero-variance prompts, we would like to direct the reviewer to the **experiments with Qwen3-8B-Base in Appendix A**, where we train on a 20k subset of **NuminaMath-CoT**. This dataset is more difficult than MATH but substantially easier than DAPO-Math-17k, resulting in a much **lower zero-variance rate - approximately 30% of prompts per batch** at the start of training (see newly added **Figure 8 in Appendix E** for precise values). We believe this experimental setup aligns closely with the reviewer's request.
>
> For convenience, we re-summarize below the evaluation results (from the manuscript) for 8B model trained on:
> - (i) the hard dataset **DAPO-Math-17k** with **lots of zero-variance prompts** (from **Table 1**)
> $$
> {\\scriptsize
> \\begin{array}{l|ll|ll|ll|ll|ll|ll}
> \\hline
> \\text{Method} &
> \\text{Minerva} & &
> \\text{AMC23} & &
> \\text{MATH500} & &
> \\text{AIME24} & &
> \\text{AIME25} & &
> \\text{OBench} & \\\\
> &
> \\text{Acc@8} & \\text{Pass@8} &
> \\text{Acc@8} & \\text{Pass@8} &
> \\text{Acc@8} & \\text{Pass@8} &
> \\text{Acc@8} & \\text{Pass@8} &
> \\text{Acc@8} & \\text{Pass@8} &
> \\text{Acc@8} & \\text{Pass@8} \\\\
> \\hline
> \\text{GRPO} &
> 44.99 & 58.42 &
> 73.75 & 92.48 &
> 83.00 & 92.40 &
> 28.33 & 50.64 &
> 20.00 & 31.59 &
> 49.59 & 66.20 \\\\
> \\hline
> \\textbf{RL-ZVP (Ours)} &
> \\mathbf{45.96} & \\mathbf{58.83} &
> \\mathbf{78.75} & \\mathbf{94.18} &
> \\mathbf{89.73} & \\mathbf{96.13} &
> \\mathbf{33.33} & \\mathbf{54.66} &
> \\mathbf{24.58} & \\mathbf{39.36} &
> \\mathbf{58.20} & \\mathbf{72.28} \\\\
> \\Delta(\\text{w.r.t GRPO}) &
> \\color[rgb]{0.2,0.6,0.2}{{(+0.96)}} & \\color[rgb]{0.2,0.6,0.2}{{(+0.41)}} &
> \\color[rgb]{0.2,0.6,0.2}{{(+5.00)}} & \\color[rgb]{0.2,0.6,0.2}{{(+1.70)}} &
> \\color[rgb]{0.2,0.6,0.2}{{(+6.73)}} & \\color[rgb]{0.2,0.6,0.2}{{(+3.73)}} &
> \\color[rgb]{0.2,0.6,0.2}{{(+5.00)}} & \\color[rgb]{0.2,0.6,0.2}{{(+4.02)}} &
> \\color[rgb]{0.2,0.6,0.2}{{(+4.58)}} & \\color[rgb]{0.2,0.6,0.2}{{(+7.77)}} &
> \\color[rgb]{0.2,0.6,0.2}{{(+8.61)}} & \\color[rgb]{0.2,0.6,0.2}{{(+6.08)}} \\\\
> \\hline
> \\end{array}
> }
> $$
>
> - (ii) the easier **NuminaMath-CoT** dataset with **fewer zero-variance prompts** (from **Table 4**).
> $$
> {\\scriptsize
> \\begin{array}{l|ll|ll|ll|ll|ll|ll}
> \\hline
> \\text{Method} &
> \\text{Minerva} & &
> \\text{AMC23} & &
> \\text{MATH500} & &
> \\text{AIME24} & &
> \\text{AIME25} & &
> \\text{OBench} & \\\\
> &
> \\text{Acc@8} & \\text{Pass@8} &
> \\text{Acc@8} & \\text{Pass@8} &
> \\text{Acc@8} & \\text{Pass@8} &
> \\text{Acc@8} & \\text{Pass@8} &
> \\text{Acc@8} & \\text{Pass@8} &
> \\text{Acc@8} & \\text{Pass@8} \\\\
> \\hline
> \\text{GRPO} &
> 42.42 & 54.41 &
> 66.88 & 90.10 &
> 83.63 & 93.32 &
> 25.00 & 45.40 &
> 20.83 & 38.49 &
> 54.01 & 68.29 \\\\
> \\hline
> \\textbf{RL-ZVP (Ours)} &
> \\mathbf{44.72} & \\mathbf{56.15} &
> \\mathbf{73.13} & \\mathbf{93.91} &
> \\mathbf{87.23} & \\mathbf{95.20} &
> \\mathbf{28.75} & \\mathbf{51.54} &
> \\mathbf{22.92} & \\mathbf{40.61} &
> \\mathbf{55.21} & \\mathbf{69.67} \\\\
> \\Delta(\\text{w.r.t GRPO}) &
> \\color[rgb]{0.2,0.6,0.2}{{(+2.30)}} & \\color[rgb]{0.2,0.6,0.2}{{(+1.74)}} &
> \\color[rgb]{0.2,0.6,0.2}{{(+6.25)}} & \\color[rgb]{0.2,0.6,0.2}{{(+3.81)}} &
> \\color[rgb]{0.2,0.6,0.2}{{(+3.60)}} & \\color[rgb]{0.2,0.6,0.2}{{(+1.88)}} &
> \\color[rgb]{0.2,0.6,0.2}{{(+3.75)}} & \\color[rgb]{0.2,0.6,0.2}{{(+6.14)}} &
> \\color[rgb]{0.2,0.6,0.2}{{(+2.09)}} & \\color[rgb]{0.2,0.6,0.2}{{(+2.12)}} &
> \\color[rgb]{0.2,0.6,0.2}{{(+1.20)}} & \\color[rgb]{0.2,0.6,0.2}{{(+1.38)}} \\\\
> \\hline
> \\end{array}
> }
> $$
>
> Across both cases, RL-ZVP consistently and substantially outperforms GRPO, confirming that our method does not harm performance in settings with fewer zero-variance prompts and in fact still provides significant gains, while yielding even larger benefits when zero-variance prompts are more prevalent.
>
> Finally, we would like to remind the reviewer of our main small-scale experiment with Qwen3-1.7B-Base on MATH, which also represents a regime with relatively few zero-variance prompts (around 25% per batch). This experiment further supports the conclusion that **RL-ZVP remains effective even when zero-variance prompts are not dominant**.

---

> ### Author Response · Authors · 2025-11-19
> **Official Response to Reviewer xC6w (3/4)**
>
> > How is it ensured that the advantages assigned to tokens in zero-variance prompts are on the same scale as normal advantages? Could differences in scale affect training stability?
>
> We use the **scaling factor $\alpha$** to reduce the magnitude of the entropy-guided advantage for tokens in zero-variance prompts, ensuring that these values remain within a range comparable to the typical GRPO-computed advantages.
>
> Regarding the question about whether differences in scale affect training stability, we kindly refer the reviewer to the **newly added experimental results examining RL-ZVP performance under different orders of magnitude for $\alpha$**. These experiments follow the same small-scale setup as our ablation study (Qwen3-1.7B-Base trained on MATH). We find that both excessively small and excessively large values of $\alpha$ harm the effectiveness of RL-ZVP: very small values underutilize the learning signals from zero-variance prompts, while very large values lead to unstable training due to overly large gradient updates from zero-variance prompts.
> $$
> {
> \\begin{array}{l|ll|ll|ll|ll|ll}
> \\hline
> \\text{Alpha} &
> \\text{Minerva} & &
> \\text{AMC23} & &
> \\text{MATH500} & &
> \\text{Avg} & \\\\
> &
> \\text{Acc@8} & \\text{Pass@8} &
> \\text{Acc@8} & \\text{Pass@8} &
> \\text{Acc@8} & \\text{Pass@8} &
> \\text{Acc@8} & \\text{Pass@8} \\\\
> \\hline
> \\alpha=0 &
> 29.09 & 43.19 &
> 42.19 & 72.00 &
> 69.09 & 84.05 &
> 46.79 & 66.41 \\\\
> \\alpha=0.001 &
> 29.41 & 43.45 &
> 43.44 & 71.08 &
> 70.65 & 86.38 &
> 47.83 & 66.97 \\\\
> \\alpha=0.01 &
> 29.09 & 43.61 &
> 43.44 & 69.75 &
> 69.77 & 86.19 &
> 47.43 & 66.52 \\\\
> \\alpha=0.1 &
> 29.96 & 44.76 &
> 48.75 & 77.19 &
> 70.98 & 87.35 &
> 49.90 & 69.77 \\\\
> \\alpha=1.0 &
> 26.98 & 41.64 &
> 39.38 & 67.30 &
> 66.60 & 84.00 &
> 44.32 & 64.31 \\\\
> \\alpha=10 &
> 24.86 & 42.01 &
> 34.69 & 65.42 &
> 60.34 & 81.97 &
> 39.96 & 63.13 \\\\
> \\hline
> \\end{array}
> }
> $$
>
> As shown in the results above, setting $\alpha$ too low reduces the performance gains over GRPO, whereas setting it too high leads to performance degradation caused by unstable optimization. **This result naturally connects to the next discussion on guidelines and heuristics for selecting an appropriate value of $\alpha$**. We appreciate the reviewer for raising this point, as it allows us to emphasize and clarify this aspect more clearly in the rebuttal and the paper.

---

> ### Author Response · Authors · 2025-11-19
> **Official Response to Reviewer xC6w (4/4)**
>
> > How should an appropriate value of $\alpha$ be chosen? Are there any guidelines or heuristics for selecting it?
>
> To select an appropriate value for $\alpha$, we first conducted a preliminary GRPO training run to measure the entropy distribution of the base model's generated tokens. Empirically, we observed that the 100th percentile of token entropy typically lies in the range [9, 11], while the 80th percentile falls around [1.5, 2.5]. Based on this observation, we choose $\alpha$ such that the scaled entropy-guided advantages fall within the same magnitude as the usual GRPO-computed advantages under the same setting. Following this principle, we selected $\alpha = 0.1$ for all experiments.
>
> We would also like to emphasize that, although $\alpha = 0.1$ was chosen using these preliminary observations, it generalizes remarkably well. RL-ZVP with this fixed scaling factor consistently and substantially **outperforms GRPO** across **two model scales** (1.7B and 8B),**three training datasets** (MATH, DAPO-Math-17k, NuminaMath-CoT), and **six evaluation benchmarks**. This strong and consistent performance demonstrates that **$\alpha = 0.1$ is robust across diverse settings** and eliminates the need for dataset or model-specific hyperparameter tuning when applying RL-ZVP in practice.

---

> ### Author Response · Authors · 2025-11-26
> **A Kind Reminder**
>
> Dear reviewer xC6w,
>
> We would like to express our sincerest gratitude for your valuable feedback, which has greatly helped us improve our work.
>
> As the rebuttal phase is coming to an end and it has been a while since our earlier responses, we would like to follow up to see whether you have any remaining concerns or questions regarding our submission. We are more than happy to continue the discussion and clarify anything needed to further strengthen the paper.
>
> Thank you once again, and we hope you have a wonderful day.

---

### Official Review · Reviewer_sAk1 · 2025-11-01

**Soundness:** 3
**Presentation:** 3
**Contribution:** 3
**Rating:** 6
**Confidence:** 4

**Summary:**

The paper targets a failure mode in RLVR training (e.g., GRPO) where prompts whose rollouts are uniformly correct or uniformly incorrect yield zero reward variance, causing normalized advantages to collapse and producing no gradient signal. The authors propose RL-ZVP, a simple modification that preserves the standard GRPO update when variance is nonzero and substitutes an entropy-guided, sign-corrected token-level advantage when variance is zero. Intuitively, positive zero-variance prompts should reinforce the behavior, while negative ones should be discouraged; using token entropy modulates the update to emphasize informative positions. Experiments on multiple math benchmarks with Qwen backbones report consistent improvements over GRPO under comparable training setups and show smoother training dynamics.

**Strengths:**

- **Well-motivated objective:** The paper clearly identifies a practical inefficiency in GRPO, zero-variance prompts producing zero gradients, then directly targets it with a minimal change to the loss. Conceptually, reclaiming learning signal from these otherwise “wasted” batches is appealing because it improves sample/compute efficiency without redesigning the overall RLVR pipeline. The proposed branch only activates when the variance is zero and leaves GRPO behavior untouched elsewhere, so the method respects the existing training regime rather than replacing it.
- **Solid empirical results:** Across several math datasets and multiple model sizes, the method reports consistent gains over GRPO under comparable setups. The improvements appear meaningful rather than marginal, and the training dynamics presented suggest more stable progress rather than spiky or brittle behavior.

**Weaknesses:**

- **Checkpoint selection leans to optimistic:** The evaluation mixes “best checkpoint during training” with other reporting choices, which can yield optimistic numbers and makes comparisons harder to interpret. It would be beneficial to follow the ML practice where checkpoints are chosen based on a held-out validation set, and results are reported from a single, consistently selected checkpoint (not a mixture).
- **Limited training-curve comparisons:** Training curves (Fig. 4) are shown only for GRPO and the proposed method. Including additional baselines would clarify whether RL-ZVP’s dynamics are uniquely beneficial or simply reflect broader trends across methods, which helps to better assess the proposal approach.

**Questions:**

see weaknesses

---

> ### Author Response · Authors · 2025-11-19
> **Official Response to Reviewer sAk1 (1/3)**
>
> We thank the reviewer for taking the time to evaluate our submission. We are glad that you find significant merits in our work, and we sincerely appreciate the constructive feedback and insightful questions, which will help us further strengthen the manuscript. We address the reviewer's concerns as follows:
>
> > **Checkpoint selection leans to optimistic**: The evaluation mixes "best checkpoint during training" with other reporting choices, which can yield optimistic numbers and makes comparisons harder to interpret. It would be beneficial to follow the ML practice where checkpoints are chosen based on a held-out validation set, and results are reported from a single, consistently selected checkpoint (not a mixture).
>
> We thank the reviewer for this comment.
>
> We would first like to clarify the rationale behind our evaluation strategy. **The mixed-evaluation approach is applied universally across all methods**, so we believe that our strategy ensured **a fair and consistent comparison between our method and the baselines**. Importantly, as the performance of trained models on different benchmarks may peak at different training steps, this strategy allows us to report the strongest results for each method on every benchmark. Although this approach is substantially more computationally expensive than evaluating a single checkpoint, we adopted it to ensure completeness and fairness across methods.
>
> Nevertheless, we fully understand the reviewer’s perspective, as single-checkpoint evaluation is an absolutely reasonable alternative. To address this concern, we report below the results of the **“single-checkpoint” version of our main table**, where we take the final model checkpoint trained with each method and evaluate it on all six benchmarks.
>
> *(To be continued in next response)*

---

> > ### Comment · Reviewer_sAk1 · 2025-11-27
> >
> > Thank you to the authors for their efforts. I still believe it's better to report results from a single checkpoint, since it's unclear whether different models benefit to varying degrees from the optimistic eval protocol.
> >
> > I appreciate the authors have now provided eval results from a single checkpoint. I will maintain my score and advocate the acceptance of this paper.

---

> > > ### Author Response · Authors · 2025-11-27
> > >
> > > Thank you for your time and effort in reviewing our paper. We are glad that the additional evaluation results addressed your concerns. We sincerely appreciate your feedback, which has been invaluable in helping us improve the manuscript.
> > >
> > > We hope you have a wonderful weekend!

---

> ### Author Response · Authors · 2025-11-19
> **Official Response to Reviewer sAk1 (2/3)**
>
> *(Continued from previous response)*
>
> **Evaluation results on small scale (Qwen3-1.7B-Base trained with MATH)**
> $$
> {\\scriptsize
> \\begin{array}{l|c|ll|ll|ll|ll|ll|ll}
> \\hline
> \\text{Method} & \\text{Num Rollouts} &
> \\text{Minerva} & &
> \\text{AMC23} & &
> \\text{MATH500} & &
> \\text{AIME24} & &
> \\text{AIME25} & &
> \\text{OBench} & \\\\
> & &
> \\text{Acc@8} & \\text{Pass@8} &
> \\text{Acc@8} & \\text{Pass@8} &
> \\text{Acc@8} & \\text{Pass@8} &
> \\text{Acc@8} & \\text{Pass@8} &
> \\text{Acc@8} & \\text{Pass@8} &
> \\text{Acc@8} & \\text{Pass@8} \\\\
> \\hline
> \\text{GRPO} &
> 172,032 &
> 28.86 & 41.62 &
> 40.00 & \\underline{69.81} &
> 68.08 & 83.20 &
> 8.75 & 17.24 &
> 4.17 & 6.67 &
> 33.20 & 50.52 \\\\
> \\text{GRPO-DS-r} &
> 172,032 &
> \\underline{29.55} & 40.55 &
> 40.00 & 59.76 &
> 66.78 & 81.96 &
> 7.50 & 17.41 &
> 2.92 & 8.73 &
> 32.67 & 49.89 \\\\
> \\textit{GRPO-DS-g} &
> 421,888 &
> 28.95 & 41.84 &
> \\underline{42.81} & 63.03 &
> \\underline{70.72} & 83.65 &
> 7.50 & \\mathbf{25.77} &
> \\mathbf{7.50} & \\underline{12.95} &
> \\mathbf{35.68} & \\underline{53.03} \\\\
> \\text{GRESO-r} &
> 173,600 &
> 27.57 & 40.33 &
> 38.44 & 60.45 &
> 67.00 & 81.83 &
> 7.08 & 18.48 &
> 3.75 & 12.89 &
> 31.30 & 49.13 \\\\
> \\textit{GRESO-g} &
> 272,032 &
> 29.14 & \\underline{42.14} &
> \mathbf{45.00} & 68.75 &
> 69.14 & \\underline{83.69} &
> \\underline{9.58} & 19.51 &
> 5.00 & 11.00 &
> 34.68 & 52.43 \\\\
> \\hline
> \\textbf{RL-ZVP (Ours)} &
> 172,032 &
> \\mathbf{29.96} & \\mathbf{43.66} &
> 42.50 & \\mathbf{71.62} &
> \\mathbf{70.98} & \\mathbf{87.35} &
> \\mathbf{12.50} & \\underline{23.99} &
> \\underline{6.25} & \\mathbf{13.23} &
> \\underline{35.11} & \\mathbf{54.84} \\\\
> \\Delta(\\text{w.r.t GRPO}) &
>  &
> \\color[rgb]{0.2,0.6,0.2}{{(+1.10)}} & \\color[rgb]{0.2,0.6,0.2}{{(+2.04)}} &
> \\color[rgb]{0.2,0.6,0.2}{{(+2.50)}} & \\color[rgb]{0.2,0.6,0.2}{{(+1.81)}} &
> \\color[rgb]{0.2,0.6,0.2}{{(+2.90)}} & \\color[rgb]{0.2,0.6,0.2}{{(+4.15)}} &
> \\color[rgb]{0.2,0.6,0.2}{{(+3.75)}} & \\color[rgb]{0.2,0.6,0.2}{{(+6.75)}} &
> \\color[rgb]{0.2,0.6,0.2}{{(+2.08)}} & \\color[rgb]{0.2,0.6,0.2}{{(+6.56)}} &
> \\color[rgb]{0.2,0.6,0.2}{{(+1.91)}} & \\color[rgb]{0.2,0.6,0.2}{{(+4.32)}} \\\\
> \\hline
> \\end{array}
> }
> $$
>
> **Evaluation results on large scale (Qwen3-8B-Base trained with DAPO-Math-17k)**
> $$
> {\\scriptsize
> \\begin{array}{l|c|ll|ll|ll|ll|ll|ll}
> \\hline
> \\text{Method} & \\text{Num Rollouts} &
> \\text{Minerva} & &
> \\text{AMC23} & &
> \\text{MATH500} & &
> \\text{AIME24} & &
> \\text{AIME25} & &
> \\text{OBench} & \\\\
> & &
> \\text{Acc@8} & \\text{Pass@8} &
> \\text{Acc@8} & \\text{Pass@8} &
> \\text{Acc@8} & \\text{Pass@8} &
> \\text{Acc@8} & \\text{Pass@8} &
> \\text{Acc@8} & \\text{Pass@8} &
> \\text{Acc@8} & \\text{Pass@8} \\\\
> \\hline
> \\text{GRPO} &
> 540,672 &
> 43.02 & 55.32 &
> 71.25 & \\underline{91.95} &
> 80.48 & 90.57 &
> \\underline{25.42} & \\underline{44.24} &
> 16.25 & 21.91 &
> \\underline{49.59} & \\underline{66.20} \\\\
> \\text{GRPO-DS-r} &
> 827,392 &
> 29.96 & 50.85 &
> 47.19 & 76.70 &
> 68.20 & 87.26 &
> 12.08 & 31.33 &
> 9.17 & 22.02 &
> 43.88 & 58.16 \\\\
> \\textit{GRPO-DS-g} &
> 2,859,008 &
> \\underline{43.24} & \\underline{57.94} &
> 70.31 & 88.67 &
> \\underline{82.90} & \\underline{91.48} &
> 18.33 & 41.80 &
> \\underline{17.08} & \\underline{27.15} &
> 44.83 & 60.42 \\\\
> \\text{GRESO-r} &
> 815,232 &
> 30.74 & 50.53 &
> 49.38 & 80.53 &
> 67.40 & 87.79 &
> 12.08 & 29.31 &
> 10.83 & 26.26 &
> 44.92 & 59.66 \\\\
> \\textit{GRESO-g} &
> 2,159,104 &
> 42.51 & 54.38 &
> \\mathbf{76.25} & 91.67 &
> 81.23 & 89.80 &
> 20.83 & 37.82 &
> 15.42 & 27.02 &
> 44.14 & 59.79 \\\\
> \\hline
> \\textbf{RL-ZVP (Ours)} &
> 540,672 &
> \\mathbf{45.96} & \\mathbf{58.83} &
> \\underline{75.00} & \\mathbf{92.50} &
> \\mathbf{89.23} & \\mathbf{95.85} &
> \\mathbf{29.17} & \\mathbf{50.41} &
> \\mathbf{22.50} & \\mathbf{35.71} &
> \\mathbf{58.20} & \\mathbf{72.28} \\\\
> \\Delta(\\text{w.r.t GRPO}) &
>  &
> \\color[rgb]{0.2,0.6,0.2}{{(+2.94)}} & \\color[rgb]{0.2,0.6,0.2}{{(+3.51)}} &
> \\color[rgb]{0.2,0.6,0.2}{{(+3.75)}} & \\color[rgb]{0.2,0.6,0.2}{{(+0.55)}} &
> \\color[rgb]{0.2,0.6,0.2}{{(+8.75)}} & \\color[rgb]{0.2,0.6,0.2}{{(+5.28)}} &
> \\color[rgb]{0.2,0.6,0.2}{{(+3.75)}} & \\color[rgb]{0.2,0.6,0.2}{{(+6.17)}} &
> \\color[rgb]{0.2,0.6,0.2}{{(+6.25)}} & \\color[rgb]{0.2,0.6,0.2}{{(+13.80)}} &
> \\color[rgb]{0.2,0.6,0.2}{{(+8.61)}} & \\color[rgb]{0.2,0.6,0.2}{{(+6.08)}} \\\\
> \\hline
> \\end{array}
> }
> $$
>
>
> RL-ZVP continues to substantially **outperform GRPO and prompt-filtering baselines**, even under disadvantageous scenarios in terms of number of rollouts.
> - RL-ZVP still **surpasses all fair baselines** with equal rollouts (GRPO, GRPO-DS-r, GRESO-r) on every benchmark.
> - For the **unfair baselines** that requires much more rollouts (GRPO-DS-g, GRESO-g), RL-ZVP remains **better on most benchmarks as well as on average**.
> - Overall, the relative ranking of methods is stable across evaluation strategies, with **RL-ZVP consistently emerging as the most effective approach**.

---

> ### Author Response · Authors · 2025-11-19
> **Official Response to Reviewer sAk1 (3/3)**
>
> > **Limited training-curve comparisons**: Training curves (Fig. 4) are shown only for GRPO and the proposed method. Including additional baselines would clarify whether RL-ZVP's dynamics are uniquely beneficial or simply reflect broader trends across methods, which helps to better assess the proposal approach.
>
> We appreciate the reviewer's suggestion to include the training dynamics of GRPO-DS and GRESO. **We have added the full training dynamics for all methods in Figure 7,  Appendix E**.
>
> We observe that at the small scale, the prompt-filtering baselines (GRPO-DS and GRESO) perform better than GRPO but remain consistently worse than our RL-ZVP method. In terms of entropy behavior, although GRPO-DS and GRESO exhibit increasing response length, their entropy rises sharply, suggesting a form of entropy explosion. In contrast, RL-ZVP achieves longer responses while maintaining stable and non-collapsing entropy, indicating a substantially healthier learning dynamic. Combined with the steady performance gains, these trends reflect the emergence of stronger long-form reasoning ability with high certainty.
>
> At the larger scale, prompt-filtering baselines lag significantly behind RL-ZVP, though they perform slightly better than GRPO. Their training dynamics also show limited growth in response length relative to RL-ZVP, and together with interior evaluation results, this suggests that GRPO-DS and GRESO fail to learn effective reasoning behaviors as efficiently as RL-ZVP.
>
> We emphasize that **this plot inherently favors GRPO-DS and GRESO**, because under the same number of gradient steps, these methods perform many more rollouts than GRPO and RL-ZVP (see Table 5). Despite this advantage for prompt-filtering baselines, RL-ZVP still shows consistent and substantial improvements over them throughout training.
>
> Finally, we would like to note that these new figures are currently placed at the end of the manuscript to avoid disrupting the figure numbering and section structure. We will reorganize and properly integrate them into the final version of the paper.

---

> ### Author Response · Authors · 2025-11-25
> **A Kind Reminder**
>
> Dear reviewer sAk1,
>
> We would like to express our sincerest gratitude for your valuable feedback, which has greatly helped us improve our work.
>
> As the rebuttal phase is coming to an end and it has been a while since our earlier responses, we would like to follow up to see whether you have any remaining concerns or questions regarding our submission. We are more than happy to continue the discussion and clarify anything needed to further strengthen the paper.
>
> Thank you once again, and we hope you have a wonderful day.

---

### Author Response · Authors · 2025-11-20
**Global Response Regarding the Revised Manuscript**

We sincerely thank all reviewers for their constructive feedback and thoughtful concerns, which have greatly helped us improve our work.

We have tried our best to address every single weakness and question raised by the reviewers. In addition, we have updated the revised manuscript by **adding a new Appendix E**, which includes the figures referenced in our rebuttal. To avoid any potential confusion for readers, we have left the main content of the paper unchanged at this stage. All additional details will be properly integrated and placed in their appropriate sections in the final camera-ready version.

Once again, we deeply appreciate your time, effort, and valuable insights. Thank you for helping us strengthen our work.

---

### Author Response · Authors · 2025-12-03
**Final Remarks by Authors**

As a final remark, we would like to sincerely thank all reviewers for their thoughtful and valuable feedback, which has greatly helped us strengthen our work. We are grateful for their recognition of our **practical and well-motivated zero-variance prompts problem in GRPO** (sAk1, 3GQ5). We also appreciate the reviewers' acknowledgement of our **strong methodology** and our **well-motivated, intuitive, and insightful solution** in RL-ZVP (all reviewers). In particular, we are encouraged that reviewers found our **comprehensive, extensive, and rigorous experimental setup impressive** - especially our comparisons not only to standard GRPO but also to two strong prompt-filtering baselines - providing **compelling evidence for RL-ZVP's superior performance and efficiency through meaningful performance improvements** (xC6w, 3GQ5). We also value the reviewers' positive remarks regarding our **in-depth empirical analysis beyond quantitative results**, including improvements in training stability through **training-dynamics analysis** (sAk1, 3GQ5) and representative **qualitative samples** (3GQ5). In addition, we are grateful to reviewer xC6w for noting that **our paper is well-written and easy to follow**.
___
During the rebuttal period, we tried our best to address the reviewers' concerns through **additional analyses and empirical evaluations**, including:
 - justification for the choice of the scaling factor $\alpha$,
 - analysis of the generalization of our chosen $\alpha$,
 - additional results and analysis on zero-variance-prompt percentages in large-scale settings with both difficult and trivial datasets,
 - full training-dynamics curves for our method and all baselines to give deeper insight into each method's learning process,
 - experiments studying how varying $\alpha$ over several orders of magnitude affects performance, and
 - justification and empirical evaluation of our asymmetric advantage design for zero-variance prompts compared to symmetric variants.
We sincerely hope that these additions address all remaining concerns regarding our work.
___
In the supplementary materials, we have also added new experimental results demonstrating the generalization of RL-ZVP. Specifically, we evaluate our method on out-of-domain tasks, including coding, general knowledge, scientific reasoning, and logical reasoning. These new results show that RL-ZVP remains effective across several reasoning tasks beyond mathematics, consistently outperforming both GRPO and strong baseline methods.
___
All additional analysis and empirical results have been incorporated into **Appendix E** of the manuscript. We hope these expanded results and discussions further highlight the contributions of our work.

Finally, we would like to express our deepest gratitude to the program chairs, area chairs, and reviewers for your efforts in hosting the conference, facilitating the discussion, and providing insightful and valuable feedback that helped us further refine our work. Thank you for your hard work, and we wish you all the best.

---

### Meta-Review · Area_Chair_8t1N · 2025-12-23

**Summary:**

Overall the reviewers found the paper well-motivated and the solution practical as demonstrated by the experiments. The paper introduces a token-level advantage-shaping strategy specifically targeting prompts with either all wrong or all correct responses (which vanilla GRPO ignores). The solution is simple but the experiments demonstrate that can be effective. Since the reviews are positive, and I believe would remain positive, I recommend acceptance. I also suggest that the authors discuss how their strategy can be combined or nor with other token level advantage shaping methods that they reference in lines 190-191.

**Reviewer Concerns:**

I believe the reviewers would maintain their scores after rebuttal. The original scores were recommending acceptance and the concerns identified were not critical and amounted mostly to additional ablation studies, which partly the authors addressed.

**Reviewer Scores:**

I believe the reviewers would maintain their scores after rebuttal

---

### Decision · Program_Chairs · 2026-01-26

Accept (Poster)